# Single Teacher, Multiple Perspectives: Teacher Knowledge Augmentation for Enhanced Knowledge Distillation

**Md Imtiaz Hossain, Sharmen Akhter, Choong Seon Hong** [*] **& Eui-Nam Huh** [*]
Department of Computer Science & Engineering, Kyung Hee University, South Korea
{hossain.imtiaz, sharmen, cshong, johnhuh}@khu.ac.kr

## Abstract

*Do diverse perspectives help students learn better?* Multi-teacher KD, which is a more effective technique than traditional single-teacher methods, supervises the student from different perspectives (i.e., teacher). While effective, multi-teacher, teacher ensemble, or teaching assistant-based approaches are computationally expensive and resource-intensive, as they require training multiple teacher networks. These concerns raise a question: *can we supervise the student with diverse perspectives using only a single teacher?* We, as the pioneer, demonstrate **TeKAP**, a novel teacher knowledge augmentation technique that generates multiple synthetic teacher knowledge by perturbing the knowledge of a single pretrained teacher i.e., **Te**acher **K**nowledge **A**ugmentation via **P**erturbation, at both the feature and logit levels. These multiple augmented teachers simulate an ensemble of models together. The student model is trained on both the actual and augmented teacher knowledge, benefiting from the diversity of an ensemble without the need to train multiple teachers. TeKAP significantly reduces training time and computational resources, making it feasible for large-scale applications and easily manageable. Experimental results demonstrate that our proposed method helps existing state-of-the-art KD techniques achieve better performance, highlighting its potential as a cost-effective alternative. Source code can be found at: https://github.com/mdimtiazh/TeKAP.

## 1 Introduction

One-hot encoded targets ($\{0, 1\}$) are very hard and rigid. Practically, achieving prediction probability similar to one-hot encoding by softmax function is not possible, as the classifier output for all the non-target classes can not be zero. Three of the major problems of a one-hot encoded system are: 1) very hard (*i.e.*, $\{0, 1\}$) which causes overfitting, 2) technically is not possible to achieve, and 3) there are no inter-class relationships information available as the target probability for all the non-target classes is same (here, $0$). Label smoothing (Müller et al., 2019) solves the first two problems where the target is changed to $\{(1 - \epsilon)/(C - 1), \epsilon\}$, which provides flexibility and reduces overfitting, where $\epsilon$ and $C$ are the softness factor (usually $\epsilon = 0.8$), and the number of classes, respectively. But similar to one-hot coding ($\{0, 1\}$), label smoothing also does not provide inter-class relationship information during training (the target probability for all the non-target classes is uniform i.e., $((1 - \epsilon)/(C - 1))$). KD first proposed by (Hinton, 2015) transfers the representational expertise of the large teacher(s) to the small student network and addresses all these three problems. The teacher logit consists of inter-class relationships (non-target class probabilities are not uniform), flexibility (target range $(0, 1)$ i.e., not $\{0, 1\}$), and practically possible to mimic the softmax output of the teacher(s).

---

*Corresponding authors.

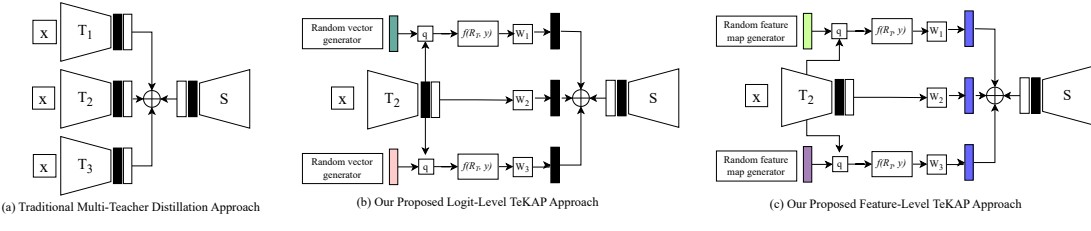

Figure 1: Depiction of the (a) multi-teacher, (b) TeKAP for logit-, and (c) feature-level distillation. $\oplus$ indicates the total of the losses after distillation (see Eq. 2, and 4). W1, W2, and W3 are the hyperparameters.

However, in KD, there are three fundamental concerns: 1) as the information comes from the teacher, is the teacher perfect for the student (Xu et al., 2020; Yang et al., 2021)? 2) is the distillation technique able to extract the teacher's intrinsic representations perfectly (Stanton et al., 2021)? and 3) the perspective of a single teacher is not diverse (as not from multiple teachers) to make the student more generalized. The first two problems are dealt with several teacher-improving (Xu et al., 2020; Yang et al., 2021), and student-friendly teacher training approaches (Park et al., 2021; Rao et al., 2023). Numerous KD approaches address the second problem (Sun et al., 2024). But, to the best of our knowledge, no prior works consider the third problem. In this paper, we focus on and investigate the third concern: *the teacher's perspective(s).* Considering teacher perspectives (Wen et al., 2024; Ma et al., 2024), and (Angarano et al., 2024; Tian et al., 2019) discussed the idea of teacher assistant, and ensemble KD, respectively, where multiple teacher knowledge is transferred to the student that enhances the student performance (Tian et al., 2019; Song et al., 2022). These approaches demand training multiple teachers, which is highly computation-heavy and resource intensive.

The discussions mentioned above led us to rethink teacher perspectives: *can we supervise the students with diverse perspectives using only single-teacher?* (Tang et al., 2020) shows that feature distortion is significantly effective in improving generalization via offering randomness where noisy labels are significantly effective to improve generalization capability further as well (Song et al., 2022). However, these works do not augment teacher knowledge and also, they do not demonstrate the potential of augmented teacher knowledge in distillation and transferability tasks. Inspired by the above discussion, we propose a novel teacher knowledge augmentation technique, **TeKAP**. TeKAP generates multiple synthetic knowledge from a pretrained teacher model through perturbation via injecting random noises at both feature and/or logit levels as shown in Fig. 1. Let's consider $f_T^i$, and $\phi_T(x)$ are the feature map from $i^{th}$ layer and teacher logit of the teacher network $\phi_T(\cdot)$, respectively, where $x$ denotes an input. Firstly, TeKAP perturbs feature map $f_T^i$ with weighted random noises $R_j^i$ to distort feature maps, where $j \in [0, 1, 3, J-1]$ and $J$ is the number of augmented teachers' knowledge to be produced. Every $j^{th}$ random noise translates the feature maps $f_T^i$ to a perturbed feature map $f_P^i$ of a different perspective. We generate $J$ number of synthetic feature maps and logits, where different $j^{th}$ noise offer diverse perspectives as discussed in section 3.1. Distorted teacher logit offers diverse inter-class relationships, flexibility, and randomness to the student. In terms of feature representations, the perturbed feature maps work as the regularization terms.

We conduct extensive experiments on standard benchmark datasets, ImageNet, CIFAR100, TinyImageNet, and STL10 for model compression, transferability, adversarial robustness, few-shot learning, scalability, and effects on occluded and noisy input tasks. The in-depth and detailed analysis and discussion demonstrate the significance of augmenting teacher knowledge. This work, TeKAP, shows a new way of knowledge representation and transfer. We argue that stochastic diverse perspectives of a single piece of information help students improve generalization.

***Why Augmented Teacher Works:*** (Allen-Zhu & Li, 2020) highlights that individual networks, if initialized

randomly, explore distinct aspects of the data, leading to diverse feature representations that enhance generalization. The dark knowledge or inter-class correlation in teacher knowledge reflects the essence of the discriminative features learned (which and what). For instance, as depicted in Fig. 2, when a model predicts a dog image as a cat or horse with a probability of $P(\text{cat}|\text{dog}) = p_c$ and $P(\text{horse}|\text{dog}) = p_h$, it indicates that the model has identified feature representations that exhibit certain similarities or dissimilarities across classes from a particular viewpoint. From a different viewpoint, the network may focus on the different sets of features. The inter-class correlation may change, for instance, $P'(\text{cat}|\text{dog}) = \hat{p}_c$ or $P'(\text{horse}|\text{dog}) = \hat{p}_h$. This variability in inter-class correlations across different viewpoints helps students learn better.

***Computational Complexity:*** TeKAP significantly reduces the computational burden associated with multi-teacher training, teaching assistant, and ensemble KD. While traditional multi-teacher-based methods require training multiple teachers, TeKAP generates multiple synthetic teacher perspectives (i.e., augmented teachers) from a single model, which helps reduce the training time and memory usage by multiple times. ***Our core contributions are:*** 1) augmenting teacher knowledge with random noise to enhance knowledge diversity for the student, as a pioneer, 2) introducing a novel method, TeKAP, for achieving diversity from a single teacher network without training, 3) demonstrating the potential of feature distortion and noisy labels in distillation and transfer, 4) presenting a plug-and-play approach that can further enhance student generalization, and 5) providing in-depth analysis and discussion on benchmark datasets for diverse vision tasks to evaluate the significance of augmented teachers through distortion.

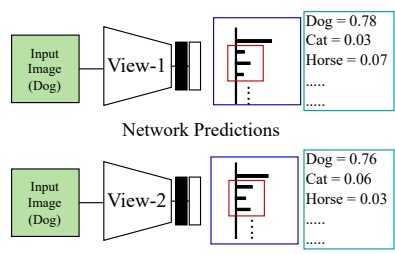

Figure 2: Depiction of an example of shifted inter-class relationship.

## 2 LITERATURE REVIEW

(Hinton, 2015) first introduced the technique *KD*. Later, many improved distillation techniques have been proposed. One very important concern in this field is *how to make the knowledge easy to learn for the student*. Several approaches have been proposed to address this problem by reducing the capacity gap between the teacher and student. (Mirzadeh et al., 2020) and (Son et al., 2021) as the seminal works exploit teacher assistants (i.e., intermediate networks) to reduce the capacity gap between the teacher and student. (Zhang et al., 2022) trained multiple teachers for improved performance. The effects of ensemble KD are explored in CRD (Tian et al., 2019) and (Allen-Zhu & Li, 2020). However, training multiple teacher networks or teacher assistants is highly resource-intensive and challenging to maintain. Another approach to achieving a better-generalized student is to improve teacher performance before distillation, such as (Xu et al., 2020), (Yang et al., 2021), or student-friendly teacher learning (Park et al., 2021; Rao et al., 2023). These approaches train auxiliary classifiers, which demand additional training of teachers, thereby increasing cost and complexity. Furthermore, these approaches generate knowledge using the same teacher output. Our work utilizes a single teacher but perturbs it using random noise, which requires no training, thereby augmenting the teacher network and achieving diversity. Another difference between existing works and TeKAP is that these approaches do not explore feature-level distortion in KD. We investigate both logits and feature-level distortion using random noise that is optimization-free. (Tang et al., 2020) shows that optimized distortion enhances network generalization and acts as a regularization term, and (Song et al., 2022) demonstrates that noisy labels generalize better. However, no one has considered teacher knowledge augmentation via perturbation, particularly for KD. One more unanswered question is *whether this distorted or noisy teacher knowledge is transferable*. We focus on demonstrating the potential of teacher perturbation and show its significance in diverse scenarios. TeKAP can be integrated with any distillation technique.

## 3 METHOD

Proposed TeKAP generates multiple augmented teachers by perturbing a single pretrained teacher model at both 1) feature, 2) logit, and 3) both levels as depicted in Fig. 1. The student is trained using the linear combination of the knowledge of both the original and augmented teachers.

### 3.1 SINGLE TEACHER, MULTIPLE PERSPECTIVES

Logit-level augmentation primarily diversifies the inter-class relationships, providing alternative supervisory signals that regularize the student network. Feature-level augmentation, on the other hand, introduces diversity in intermediate feature representations, exposing the student to a broader spectrum of variations (like dropout or data augmentation). Both augmentations target distinct aspects of teacher knowledge: logits focus on prediction diversity, while features address internal representation diversity.

**Feature-level Perturbation:** Feature perturbation introduces diversity into the intermediate feature representations of the teacher network $\phi_T$, allowing the student network $\phi_S$ to learn from diverse perspectives. Let, $f_T(x) \in \mathbb{R}^{h \times w \times c}$ be the feature map generated by the teacher for an input instance $x$, where $h, w$, and $c$ represent the height, width, and number of channels, respectively. To augment diverse teacher perspectives, we perturb the teacher's feature map with Gaussian noise $\eta_i \sim \mathcal{N}(0, \sigma^2)$ to generate synthetic feature maps:

$$f_T^{(i)}(x) = \alpha \times \eta_i + (1 - \alpha) \times f_T(x) \tag{1}$$

Here $\alpha = 0.1$ is a scaling factor controlling the perturbation intensity. We have used zero mean and 1 std. to produce random noise on every epoch and perform a weighted combination with the original teacher logits (noise weights with 0.1 and teacher weights with 0.9). This results in a set of perturbed feature maps that provide alternate augmented teacher outputs. The student is trained using both the original and synthetic feature maps through a feature distillation loss, such as:

$$\mathcal{L}_{feat} = \lambda \mathrm{L}(f_S(x), f_T(x)) + (1 - \lambda) \sum_{i=1}^{N} \mathrm{L}(f_S(x), f_T^i(x)) \tag{2}$$

Here $f_S(x)$ is the student feature map and $\lambda$ controls the balance between the original and synthetic knowledge transfer using distillation loss $\mathrm{L}(\cdot, \cdot)$.

**Justification for Feature-level Perturbation:** *Feature perturbation can be viewed as a form of regularization, similar to dropout, or data augmentation, which helps models generalize better (Tang et al., 2020). Perturbing the feature maps exposes the student to a range of variations around the original feature map. When a model is exposed to multiple noisy versions of the same input, it is forced to learn a more robust inductive bias i.e., mapping without being overconfident (Allen-Zhu & Li, 2020).*

**Logit-Level Perturbation:** The idea is to diversify the teacher's output logits to provide multiple synthetic supervision signals to the student at the logit level along with feature-level perturbation. Let $z_T(x)$ represent the logits (post-softmax activations) produced by the teacher network $\phi_T$ for an input $x$, where $z_T(x) \in \mathbb{R}^C$ and $C$ is the number of classes. We introduce a perturbation $\eta_i \sim \mathcal{N}(0, \sigma^2)$ sampled from a Gaussian distribution, and add it to the teacher's logits:

$$z_T^{(i)}(x) = \alpha \times \eta_i + (1 - \alpha) \times z_T(x) \tag{3}$$

Here, we use $\alpha = 0.1$. These perturbed logits $z_T^{(i)}(x)$ represent the augmented version of the teacher prediction. The student network $\phi_S$ is trained using both of the original, $z_T(x)$, and synthetic logits $z_T^{(i)}(x)$.

$$\mathcal{L}_{logits}^{\text{perturb}} = \lambda \mathcal{L}_{KD}(z_S(x), z_T(x)) + (1 - \lambda) \sum_{i=1}^{N} \mathcal{L}_{KD}(z_S(x), z_T^{(i)}(x)) \tag{4}$$

where $\mathcal{L}_{KD}$ is the distillation loss, and $\lambda$ is the balancing weights between the original and synthetic logits.

**Justification for Logit-level Perturbation:** *These noisy logits act as different perspectives of the teacher's prediction. The inter-class relationships or dark knowledge is transformed into different sets of combinations that help the student to be regularised better (Song et al., 2022). The student, instead of overfitting to a single set of logits, is now forced to generalise across multiple noisy versions. This corresponds to learning a broader range of decision boundaries, making the student network more robust.*

**Unified Framework for Knowledge Transfer:** The student model $\phi_S$ is trained using both the original teacher outputs (feature map: $f_T(x)$, logits: $z_T(x)$), and the synthetic perturbed outputs (feature maps: $f_T^{(i)}(x) = (1-\alpha)f_T(x) + \alpha\eta_i$, logits: $z_T^{(i)}(x) = (1-\alpha)z_T(x) + \alpha\eta_i$). TeKAP combines both *logit* and *feature-level* perturbations during training student:

$$\mathcal{L}_{TeKAP} = \gamma_1 \times \mathcal{L}_{feat} + \gamma_2 \times \mathcal{L}_{logit} + \gamma_3 \times \mathcal{L}_{cel} \tag{5}$$

Here, $\gamma_1$, $\gamma_2$, and $\gamma_3$ are the balancing weights which can be adopted from the corresponding distillation technique. $\mathcal{L}_{feat}$, $\mathcal{L}_{logit}$, and $\mathcal{L}_{cel}$ represent feature level, logits level and cross-entropy loss, respectively. As the representative approach, we adopt CRD (Tian et al., 2019) as the feature-level and KD (Hinton, 2015) as the logit-level distillation techniques. We use $\gamma_1 = 0.8$, $\gamma_2 = 0.2$, and $\gamma_3 = 1$ by adopting the weights from CRD (Tian et al., 2019) for fair comparison.

**Dynamic Noise Perturbation** Injected Gaussian noise into both logits and feature maps are refreshed at each batch. This continuous noise variation ensures extra diversity for the student. The resulting noise-driven diversity improves the student's generalization performance by simulating a dynamic ensemble of teacher perspectives throughout training.

## 3.2 THEORETICAL PROOF

In TeKAP, the gradient of the loss function with respect to the student parameters $w_S$ becomes:

$$\nabla_{w_S} \mathcal{L}_{feat}^{\text{perturb}} = \lambda \nabla_{w_S} \text{L}(f_S(x), f_T(x)) + (1-\lambda) \sum_{i=1}^{N} \nabla_{w_S} \text{L}(f_S(x), f_T^i(x)) \tag{6}$$

where, $\nabla_{w_S}$, $L_{\text{feat}}$, $f_S(x)$, $f_T(x)$, $f_T^i(x)$, and $\lambda$ represent the gradient of the loss function for the student network's parameters $w_S$, feature-level distillation loss function, feature map output of the student, feature map output of the teacher, perturbed version of the teacher's feature map, and a weighting factor that balances the contributions of the original teacher knowledge and the perturbed knowledge in the loss function, respectively. Since $f_T^{(i)}(x)$ adds variation to the teacher's representation, this noise helps the student generalize better to unseen data, as it's exposed to a wider set of teachers' perspectives. Adding noise smooths the loss surface, which helps students in optimization landscapes find better solutions with a lower generalization error. This is analogous to data augmentation or dropout methods. The expected loss in the original teacher:

$$\mathbb{E}_{x \sim \mathcal{X}}[\mathcal{L}_{feat}] = \mathbb{E}_{x \sim \mathcal{X}} \text{L}(f_S(x), f_T(x)) \tag{7}$$

and in the perturbed case:

$$\mathbb{E}_{x \sim \mathcal{X}} \left[ \mathcal{L}_{feat}^{\text{perturb}} \right] = \lambda \mathbb{E}_{x \sim \mathcal{X}} \text{L}(f_S(x), f_T(x)) + (1-\lambda) \sum_{i=1}^{N} \mathbb{E}_{x \sim \mathcal{X}} \text{L}(f_S(x), f_T^i(x)) \tag{8}$$

where N represents the number of augmented (perturbed) teachers. The discussion above indicates that since the perturbed features $f_T^{(i)}(x)$ are different noisy versions of the same feature map, the second term will generally have higher variability. Let's $\mathcal{T}(x)$ denote the knowledge of the teacher model for an input $x$, and $\tilde{\mathcal{T}}_k(x) = (1-\alpha)\mathcal{T}(x) + \alpha\epsilon_k$ be the noisy prediction where $\epsilon_k \sim \mathcal{N}(0, \sigma^2)$ represents Gaussian noise added

to the teacher's output. The diversity introduced by Gaussian noise can be analyzed using Rademacher complexity (discussed in (Hsu et al., 2021), and (Tang et al., 2020)), which measures the capacity of the hypothesis class to fit random noise. Let $\mathcal{H}$ be the hypothesis class for the student model $\mathcal{S}$. The Rademacher complexity $\hat{\mathcal{R}}_n(\cdot)$ of $\mathcal{H}$ with respect to noisy predictions $\tilde{\mathcal{T}}_k(x)$ is given by:

$$\hat{\mathcal{R}}_n(\mathcal{H}) = \frac{1}{n}\mathbb{E}_\sigma \left[ \sup_{h \in \mathcal{H}} \sum_{i=1}^n \sigma_i h(x_i) \right] \tag{9}$$

where $\sigma_i$ are Rademacher variables and sup stands for supremum. h and n represent a function from the hypothesis class $\mathcal{H}$ and the number of training samples, respectively. The addition of noise $\epsilon_k$ increases the variability of predictions:

$$\mathrm{Var}[\tilde{\mathcal{T}}_k(x)] = \mathrm{Var}[\mathcal{T}(x) + \epsilon_k] = \mathrm{Var}[\mathcal{T}(x)] + \sigma^2 \tag{10}$$

where $\sigma^2$ indicates the variance of the Gaussian noise which determines the spread or intensity of the perturbation. This increased variability enhances the Rademacher complexity:

$$\hat{\mathcal{R}}_n(\mathcal{H}) = \frac{1}{n}\mathbb{E}_\sigma \left[ \sup_{h \in \mathcal{H}} \sum_{i=1}^n \sigma_i (\mathcal{T}(x_i) + \epsilon_i) \right] \tag{11}$$

Thus, the student model trained with noisy predictions benefits from increased diversity, which can reduce generalization error by improving the fit of the training data. The empirical risk of the student model $\mathcal{S}$ trained with noisy predictions is:

$$\hat{\mathcal{L}}_{\mathrm{emp}}(\mathcal{S}) = \frac{1}{n} \sum_{i=1}^n \mathrm{KL}(p(\tilde{\mathcal{T}}_k(x_i)) \| p(S(x_i))) \tag{12}$$

where $\hat{\mathcal{L}}_{\mathrm{emp}}$ and $KL(\cdot)$ indicate empirical loss and KL divergence, respectively. The *Generalization Error (GE)* bound is:

$$\mathrm{GE} \leq \hat{\mathcal{L}}_{\mathrm{emp}}(S) + \sqrt{\frac{2\hat{\mathcal{R}}_n(\mathcal{H})^2 \log(2/\delta)}{n}} \tag{13}$$

where $\delta$ is a confidence parameter. The noise addition helps to achieve lower empirical risk and better generalization by increasing the complexity as regularization terms with diverse perspectives. Extended theoretical analysis can be found in the supplementary.

| | To Similar Architecture | | | | | To Different Architecture | | |
|---|---|---|---|---|---|---|---|---|
| Teacher
Student | resnet32x4
resnet8x4 | WRN_40_2
WRN_40_1 | WRN_40_2
WRN_16_2 | VGG13
VGG8 | resnet56
resnet20 | resnet32x4
ShuffleNetV1 | resnet32x4
ShuffleNetV2 | WRN-40-2
ShuffleNetV1 |
| Teacher
Student | 79.42
72.50 | 75.61
71.98 | 75.61
73.26 | 74.64
70.36 | 72.34
69.06 | 79.42
70.50 | 74.64
70.36 | 75.61
70.50 |
| KD
**+ TeKAP (L)** | 73.33
**74.79** | 73.69
**73.80** | 74.92
**75.21** | 72.98
**74.00** | 70.66
**71.32** | 74.07
**74.92** | 72.98
**75.43** | 74.83
**76.75** |
| CRD
**+ TeKAP (F)** | 75.51
**75.65** | 74.14
**74.21** | 75.48
**75.83** | 73.94
**74.10** | 71.16
**71.71** | 75.11
**75.55** | 75.65
**76.23** | 76.05
**76.60** |
| **TeKAP (F+L)** | 75.98 | 74.41 | 76.20 | 74.42 | 71.92 | 75.60 | 77.38 | 76.59 |

Table 1: The effects of TeKAP on the SOTA methods. Competing results and setups for KD and CRD are quoted from CRD ((Tian et al., 2019)). F and L indicate feature and logit-level distortions, respectively.

# 4 EXPERIMENTS

We have discussed the detailed results and analysis in this section. The rest of the results on the complexity, class imbalance datasets, noise sensitivity, hyperparameters analysis, etc., are available in the supplementary.

## 4.1 EFFECT OF TEKAP ON RECENT SOTAS.

| Baselines | Teacher
Student | resnet32x4
resnet8x4 | WRN_40_2
WRN_40_1 |
|---|---|---|---|
| | Teacher
Student | 79.42
72.50 | 75.61
71.98 |
| Single Teacher | DKD
+ TeKAP | 76.32
**76.59** | 74.81
**75.33** |
| | MLKD
+ TeKAP | 77.08
**77.36** | 75.35
**75.67** |
| Multi- Teacher | TAKD
+ TeKAP | 73.93
**74.81** | 73.83
**74.37** |
| | CA-MKD
+ TeKAP | 75.90
**76.34** | 74.56
**74.98** |
| | DGKD
+ TeKAP | 75.31
**76.17** | 74.23
**75.14** |

Table 2: Effects of TeKAP on the SOTAs.

Table 2 illustrates the performance impact of incorporating TeKAP into several state-of-the-art (SOTA) KD methods, including DKD (Zhao et al., 2022), MLKD (Jin et al., 2023), TAKD (Mirzadeh et al., 2020), CA-MKD (Zhang et al., 2022) and DGKD (Son et al., 2021). The experiments are conducted using two teacher-student model setups: (1) resnet32x4 as the teacher and resnet8x4 as the student, and (2) WRN_40_2 as the teacher and WRN_40_1 as the student on CIFA100 dataset. For all methods, the inclusion of TeKAP leads to consistent performance improvements, as shown by the higher accuracy values for students trained with TeKAP-enhanced methods. DKD shows an improvement from 76.32% to 76.59% for resnet32x4-resnet8x4 and from 74.81% to 75.33% for WRN_40_2-WRN_40_1. MLKD gains similarly, with improvements reaching 77.36% and 75.67%, respectively. TeKAP significantly enhances TAKD, boosting it from 73.93% to 74.81% and from 73.83% to 74.37%, highlighting its ability to bridge capacity gaps. For CA-MKD and DGKD, TeKAP integration results in moderate yet meaningful gains, affirming its compatibility with advanced multi-teacher and dynamic distillation strategies. This table underscores the versatility and effectiveness of TeKAP as a plug-and-play augmentation technique that consistently elevates the generalization performance of various distillation approaches.

## 4.2 SIGNIFICANCE REGARDING LOGIT AND FEATURE LEVEL METHODS

**Performance on CIFAR100:** Table 1 shows the effect of TeKAP on SOTA logits-based (KD (Hinton, 2015)), and feature-based (CRD (Tian et al., 2019)) knowledge distillation approaches for similar and different teacher-student architecture setups on the CIFAR100 dataset. The table shows that TeKAP uplifts the performance of KD and CRD in all the scenarios. In some cases, the performance gains are significant, such as (WRN_40_2-ShuffleNetV1), and (resnet32x4-ShuffleNetV2). In the case of resnet20, and WRN_40_1, the performance gain is nominal. This is because the student resnet20 and WRN_40_1 are tiny.

In some cases, our approach TeKAP helps *KD to beat even the teacher networks* such as WRN-40-2-ShuffleNetV1, and resnet32x4-ShuffleNetV2 setup, where TeKAP (L) helps KD achieve 1.14%, and 0.79% improved accuracy *than original teacher*. In the case of (WRN_40_2-ShuffleNetV1), TeKAP helps KD to achieve better performance than CRD. The performance gains by TeKAP(F+L) verify that

| Set | Teacher | Student | KD | KD + TeKAP (L) |
|---|---|---|---|---|
| Top-1 | 26.69 | 30.25 | 29.59 | **29.33** |
| Top-5 | 8.58 | 10.93 | 10.30 | **10.08** |

Table 3: Scalability of TeKAP on ImageNet dataset.

both the distorted feature and logits are transferable to the student. TeKAP(T+L) indicates both feature and logit level distortions achieve significant performance improvements. The results suggest that if the student is capable and the distillation approach is effective, then the diversity of the teacher and shifting inter-class correlation help the student improve performance.

**ImageNet-1K: Scalability on Large-Scale Dataset:** We evaluate the scalability and show the performance comparison on the large-scale ImageNet dataset in Table 3. The teacher-student architecture setups and competing results are adopted from CRD (Tian et al., 2019) where ResNet-34 and ResNet-18 are considered as the teacher and student, respectively. From Table 3, we notice that TeKAP helps KD achieve improved accuracy for both top-1 and top-5 error rates. These performance improvements verify the scalability of the proposed TeKAP in large-scale datasets. Shifting inter-class correlations and diversity in dark knowledge offers discriminative feature representations which makes the representation easy to learn for the student.

### 4.3 TeKAP vs Multiple Teacher assistant based KD approach TAKD

| Teacher
Student | resnet32x4
resnet8x4 | WRN_40_2
WRN_40_1 | WRN_40_2
WRN_16_2 | VGG13
VGG8 | resnet56
resnet20 | resnet32x4
ShuffleNetV1 | resnet32x4
ShuffleNetV2 | WRN-40-2
ShuffleNetV1 |
|---|---|---|---|---|---|---|---|---|
| TAKD | 73.81 | 73.78 | 75.12 | 73.23 | 70.83 | 74.53 | 74.82 | 75.34 |
| **TeKAP (L)** | **74.79** | **73.80** | **75.21** | **74.00** | **71.32** | **74.92** | **75.43** | **76.75** |
| **TeKAP (F+L)** | 75.98 | 74.41 | 76.20 | 74.42 | 71.92 | 75.60 | 77.38 | 76.59 |

Table 4: Performance comparison between TeKAP and multi-teacher approach TAKD.

TeKAP using a single pretrained teacher and without training any additional network except student beats the performance of SOTA teaching assistant-based approach TAKD (Mirzadeh et al., 2020) in all the scenarios (Table 4). TAKD reduces teacher-student capacity gaps by training multiple teaching assistants, which is computationally expensive. On the other hand, using a single teacher and an ignorable computation (generating noise and calculating loss) TeKAP beats TAKD. *This improvement depicts that augmented diversity also reduces the teacher-student capacity gap* while providing diversity and reducing training complexity. This gain indicates that the augmented diversity reduces the teacher-student capacity gap better than TAKD.

### 4.4 Effect of TeKAP on class imbalance datasets.

Table 5 shows the effectiveness of TeKAP(L) in addressing class imbalanceness in KD tasks on CIFAR100 dataset. TeKAP improves the performance of all three teacher-student pairs over the baseline approaches. Detailed analysis can be found in the supplementary.

| Methods | resnet32x4-resnet8x4 | WRN_40_2-WRN_16_2 | VGG13-VGG8 |
|---|---|---|---|
| Baseline (KD) | 41.71 | 52.08 | 47.52 |
| + TeKAP (Ours) | 46.42 | 52.72 | 51.25 |

Table 5: TeKAP on class imbalance dataset (CIFAR100).

### 4.5 Inter-Class Correlations Comparison

Inter-class correlation is the structural knowledge of a teacher consisting of which and what discriminative features have been learned by a network. Diversity in the predictions helps students generalize better. Fig. 3 shows the comparisons regarding the inter-class correlation differences of the vanilla student, baseline KD, and TeKAP. We used resnet32x4 and resnet8x4 as the teacher-student setups and trained on the CIFAR100 dataset. The differences in inter-class correlation between the

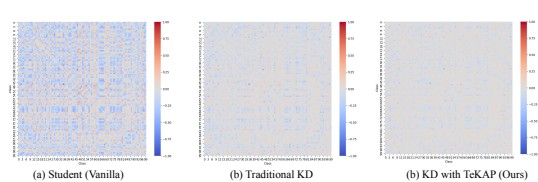

(a) Student (Vanilla)   (b) Traditional KD   (b) KD with TeKAP (Ours)

Figure 3: Comparison on inter-class correlations.

teacher and TeKAP students are lower compared to baseline vanilla and KD students. Surprisingly, though

the augmented teacher adds diversity, this diversity or distortion helps students achieve teacher inter-class correlation more perfectly than baseline KD which further justifies the claims in (Tang et al., 2020) and (Sun et al., 2024). The diversity of the augmented teachers works as the regularization terms.

| Set | Student | KD | KD + TeKAP (L) |
|---|---|---|---|
| CIFAR100-STL10 | 70.33 | 71.01 | **72.94** |
| CIFAR100-TinyImageNet | 34.82 | 35.53 | **35.81** |

| | Teacher | Student | KD | KD + TeKAP (L) |
|---|---|---|---|---|
| top-1 | 29.88 | 21.44 | 21.12 | **22.47** |
| top-5 | 51.43 | 44.47 | 44.04 | **45.72** |

Table 6: Transferability: The learned representation on the CIFAR100 is transferred to STL-10 and TinyImageNet.

Table 7: Adversarial robustness of our proposed TeKAP on CIFAR100.

### 4.6 TRANSFERABILITY TO DIFFERENT DATASETS

The student trained by TeKAP achieves better transferability to different datasets (TinyImageNet and STL10) than the baseline (Table 6). The representation learned from the CIFAR100 is transferred to STL10 and TinyImageNet datasets. The network is finetuned to STL10 and TinyImageNet datasets where (resnet32x4 - resnet8x4) is considered as the teacher-student setup. TeKAP-student achieves $1.93\%$, and $0.28\%$ higher accuracy than KD on CIFAR100-STL10, and CIFAR100-TinyImageNet transfer setup, respectively. The diversity induced by the augmented teachers enhances student transferability to different datasets.

### 4.7 EFFECT ON FEW-SHOT LEARNING SCENARIOS: LEARNING FROM A SMALL DATASET

TeKAP transfers multiple augmented diversity to the student using a single input. Using TeKAP, a student can learn more variations from a smaller number of data compared to the baselines. We have evaluated TeKAP on the few-shot learning task. The effect of the augmented teachers is evaluated on $25\%$, $50\%$, and $75\%$ of the training set using resnet32x4-resnet8x4 teacher-student setup. The students are trained using a portion of the dataset. TeKAP improves performance in every scenario as shown in Fig. 4. These results demonstrate that TeKAP, as an augmentation technique, effectively provides many variations and helps to learn better from the small amount of data.

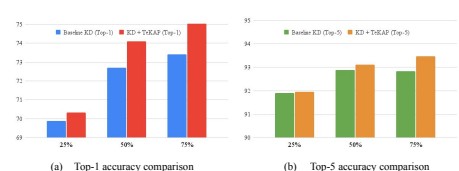

(a) Top-1 accuracy comparison    (b) Top-5 accuracy comparison

Figure 4: Effects of TeKAP on the few shot training scenarios ($25\%$, $50\%$, and $75\%$ data of the CIFAR100).

### 4.8 ADVERSARIAL ROBUSTNESS

TeKAP offers diverse and random dark knowledge of inter-class correlations. The networks learn variations that make a student robust against adversarial attacks compared to the baselines (Table 7). TeKAP uplifts the performance of KD by an improved accuracy of $1.35\%$ on the CIFAR100 in resnet32x4-resnet8x4 teacher-student setup. During training, both the teacher and student use a clean train set, while during the evaluation of the student, we attack the test set using the FGSM adversarial attack method where $\epsilon = 0.005$ (Madry et al., 2017). As the network learns noisy and random variations it becomes robust to adversarial attacks.

### 4.9 COMPARATIVE COMPUTATIONAL COMPLEXITY

The training of a teacher ResNet32x4 in CIFAR100 using KD takes approximately 16 seconds per epoch. For 240 epochs, the total time taken is $240 \times 16 = 64$ minutes using two 3080 NVIDIA GeForce GPUs. For multi-teacher or ensemble learning, we need to train multiple teachers. Let's assume two teacher assistants of equal size, which takes $64 \times 2 = 128$ minutes (approx) for DGKD Son et al. (2021). In our approach, TeKAP takes 18 seconds per epoch, which is 72 minutes in total. TeKAP avoids training multiple teachers.

## 4.10 Effects on Patch Occlusion i.e., Noisy Data

|       | Teacher | Student | KD    | KD + TeKAP (L) |
|-------|---------|---------|-------|----------------|
| top-1 | 76.16   | 71.79   | 71.85 | **72.85**      |
| top-5 | 92.86   | 92.27   | 92.08 | **92.57**      |

Table 8: Performance on noisy data. A random noisy patch is mixed at a random position.

TeKAP provides added variations i.e., diversity to the student during training that makes the student more robust to unseen variations and noises. We evaluated the robustness of the TeKAP against occlusion and noise on the CIFAR100 dataset using resnet32x4-resnet8x4 teacher-student setup as shown in Table 8. Both the teacher and student are trained using a clean dataset where the noise in input is added during evaluations of the student. We add a random patch of $(4 \times 4)$ dimensions at random positions of the input images for every batch. From Table 8 it is observed that the students trained with augmented teacher perform better in noisy data which further verifies the significance of augmented diversity.

## 4.11 Effects of TeKAP on Ensemble Learning

| # Teachers | T=1   | T=2   | T=3   |
|------------|-------|-------|-------|
| Accuracy   | 75.98 | 76.12 | 76.19 |

Table 9: TeKAP on ensemble learning.

Fig. 5 shows the effect of the number of augmented teachers. We use resnet32x4-resnet8x4 as the teacher-student setup on the CIFAR100 dataset to examine the effect of the hyper-parameters. From Fig. 5 we see that TeKAP is robust to the number of augmented teachers. For every number of augmented teachers, TeKAP achieves better accuracy than the baseline. The accuracy improves as the number of augmented teachers increases. However, this experiment is run with up to ten augmented teachers.

Table 9 shows the effect on ensemble learning. Three (3) augmented teachers are used for every original teacher. During feature and logit distortion, the weights for noise and teacher output are 0.1, and 0.9, respectively.

## 5 Conclusion

This paper proposes a novel and innovative teacher knowledge augmentation framework that offers multi-teacher diversity using a single-teacher model by augmenting teacher prediction with random noise. As pioneers, we show that both feature and logit level distortion or noisy predictions are transferable to the student. Our work, TeKAP, dynamically generates synthetic knowledge that helps students improve generalization. We demonstrate extensive evaluation with discussions on model compression, scalability, transferability, few-shot learning, adversarial robustness, and the effect of noisy data. The proposed TeKAP improves the performance of the existing KD approach on benchmark datasets. Our proposed TeKAP offers diverse knowledge using a single teacher and avoids training multiple teachers. The augmented diversity also reduces the teacher-student capacity gap. We do not propose any new KD

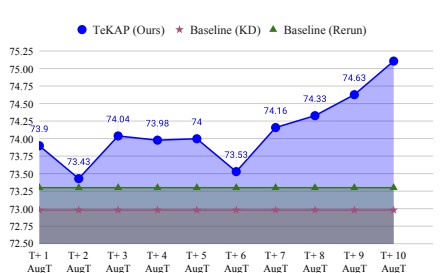

Figure 5: Effects of the number of augmented teachers on TeKAP on KD

technique. TeKAP can be easily applicable to any existing KD approach. This paper opens up a new research direction for teacher knowledge augmentation and achieving diversity using a single teacher.

**Limitations and Future work** The proposed TeKAP does not optimize the noise and remains train-free. Our proposed TeKAP enjoys the benefits of randomness and diversity. For future work, we plan to explore optimization-based techniques for teacher knowledge distortion.

ACKNOWLEDGMENTS

This work was partly supported by the Institute of Information & Communications Technology Planning & Evaluation(IITP)-ITRC(Information Technology Research Center) grant funded by the Korea government(MSIT) (IITP-2025-RS-2023-00258649, 50%) and the Institute of Information and Communications Technology Planning and Evaluation (IITP) grant funded by the Korea Government (MSIT) (RS-2021-II212068, Artificial Intelligence Innovation Hub, 50%).

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
