# OpenReview forum: "Single Teacher, Multiple Perspectives: Teacher Knowledge Augmentation for Enhanced Knowledge Distillation"
_ICLR.cc/2025/Conference — ICLR 2025 Poster_

### Official Review · Reviewer_FGoU · 2024-11-01

**Soundness:** 3
**Presentation:** 3
**Contribution:** 3
**Rating:** 6
**Confidence:** 3

**Summary:**

The authors propose TeKAP, a novel teacher knowledge augmentation technique that generates diverse synthetic teacher knowledge by perturbing a single pretrained teacher. This plug-and-play module leverages simple perturbations to capture ensemble benefits without training multiple teachers. Experimental results demonstrate TeKAP's effectiveness in enhancing both logit and feature-based knowledge distillation methods.

**Strengths:**

- The proposed plug-and-play module integrates seamlessly with existing KD methods, adding minimal computational burden.
- By augmenting knowledge from a single pretrained teacher network, the authors significantly reduce training time and resource demands while achieving ensemble-like effects.
- The approach is simple yet highly effective.

**Weaknesses:**

- The proposed plug-and-play module was not well validated. Specifically, it was only applied to vanilla KD and CRD, even though there have been many advanced KD methods that can serve as baselines.
- The experiments omit numerous state-of-the-art single-teacher and multi-teacher KD methods; additional benchmark comparisons would - strengthen the evaluation.
- Details on dynamic noise perturbation are insufficient, with critical implementation information missing for reference.

**Questions:**

-How can randomly distorted teacher logits provide diverse inter-class relationships if the distortion is truly random?
-What does h represent in Eq. 9?
-What is the scale of the random noise, and how should it be set? Detailed guidelines for noise settings are needed.
-There appears to be no discernible difference between Fig. 3(b) and Fig. 3(c).

---

> ### Author Response · Authors · 2024-11-24
> **Response to Reviewer FGoU (1/2)**
>
> Thank you for your constructive feedback. We appreciate the insights from the reviewers. Below, we address each of the points step-by-step:
> ### **Additional Experiments:**
>
> |        | Model         | ResNet32x4-ResNet8x4 | WRN_40_2-WRN_40_1 |
> |-----------------|---------------|------------|----------|
> | **Teacher**     | Accuracy      | 79.42      | 75.61    |
> | **Student**     | Accuracy      | 72.50      | 71.98    |
> | **Single Teacher** | DKD [1]   | 76.32      | 74.81    |
> |                 | **DKD + TeKAP (Ours)**   | **76.59**  | **75.33**|
> |                 | MLKD [2]     | 77.08      | 75.35    |
> |                 | **MLKD + TeKAP (Ours)**   | **77.36**  | **75.67**|
> |  **Multi-Teacher** | TAKD [3]      | 73.93      | 73.83    |
> |                 | **TAKD + TeKAP (Ours)**   | **74.81**  | **74.37**|
> |                 | CA-MKD [4] | 75.90      | 74.56    |
> |                 | **CA-MKD + TeKAP (Ours)**   | **76.34**  | **74.98**|
> |                 | DGKD [5]    | 75.31      | 74.23    |
> |                 | **DGKD + TeKAP (Ours)**   | **76.17**  | **75.14**|
>
>
> **Table-1:** The effects of TeKAP on the SOTA methods DKD [1], MLD [2], TAKD [3], CA-MKD [4], and DGKD [5].
>
>
> | #Original Teachers (T) | TeKAP (Ours) |
> |-------------------------|--------------|
> | 1 OriginT + 3 AugT             | 75.98        |
> | 2 OriginT + 3 AugT             | 76.12        |
> | 3 OriginT + 3 AugT             | 76.31        |
>
> Table 2: Effect of multiple original teachers.
>
> |Network |  Augmentation Techniques | TeKAP (F+L) |
> |-------------------------|--------------|--------------|
> |     **ResNet32x4-ResNet8x4**         | Gaussian             | 75.98        |
> |                                                             | Uniform             | 75.71        |
> |     **WRN_40_2-WRN_40_1**         | Gaussian             | 74.41        |
> |                                                             | Uniform             | 74.26        |
>
> Table 3: Effect of different noise techniques.
>
> ###  **Responses**:
> 1. **Validation of the Proposed Module:** We have conducted additional experiments comparing TeKAP with several state-of-the-art single-teacher and multi-teacher KD methods in Table 1. These results, which include advanced methods like DKD[1], MLKD[2], TAKD[3], CA-MKD[4], and DGKD[5]. Also, we have added additional evaluations in terms of ensemble, multi-teacher augmentations and so on in Tables 2 and 3.
>
>
> 2. **Details on Dynamic Noise Perturbation:** The generation of random noise happens on every epoch which we addressed as dynamic noise perturbation.
>
> 3. **Scale of random noise:** We have used zero mean and 1 std. to produce random noise on every epoch and perform a weighted combination with the original teacher logits (noise weights with 0.1 and teacher weights with 0.9). We also added detailed guidelines on how to set the noise parameters for different configurations.
>
> 4. **Meaning of h in Eq 9:** h represents a function from the hypothesis class H, which is a set of functions under consideration. Each h maps inputs x_i​ (from the dataset) to real numbers, often representing predictions, scores, or decisions. This measures the capacity or complexity of H.
>
> 5. **Clarification of Random Distortion and Inter-Class Relationships:** While the noise is random, it serves to introduce variability that prevents the student from overfitting to the teacher’s exact logits (i.e. single perspectives). If two classes are strongly correlated in the teacher logits, random distortions will not eliminate this correlation but may perturb its exact magnitude or direction, leading to diverse interpretations of the relationship. Imagine teaching a concept by showing slightly varied examples, this helps learners generalize the concept rather than memorize specific instances. Similar to techniques like dropout (which can be considered implicitly network ensemble learning because every random dropping creates a different network structure), random feature distortion (considered as a diverse network as the outputs are slightly different so it is assumed they come from different networks) can force the model to adapt to a broader range of conditions. This diversity helps the student model avoid collapsing into a rigid interpretation of the teacher’s outputs.
>
> 4. **Fig. 3(b) vs Fig. 3(c):** Thanks for pointing this out. Usually in knowledge distillation feature and logits level knowledge distillation are divided into different categories. So to express that our approach is applicable to both the feature and logits level, we have drawn both feature and logits level figure. Actually, the figure describes that our TeKAP is a applicable to both feature 3(b) and logit level 3(c).
>
>
> We hope these revisions address your concerns and improve the clarity of the paper. We believe that these additional experiments and clarifications strengthen our work. Thank you again for your valuable feedback.

---

> ### Author Response · Authors · 2024-11-24
> **Response to Reviewer (FGoU) (2/2)**
>
> ### The improvements we have made:
> 1. **(Reviewers: NkEk, pUYy, zdP5, FGoU): Additional comparison with state-of-the-art:** Added to the revised manuscript (Table 2, page 7)
>
> 2. **(Reviewers: NkEk, pUYy, zdP5, FGoU) multi-teacher:** The results discussion for the recent SOTA multi-teacher approach is added to section 4.1, Table 2 (page 7) of the revised manuscript.
>
> 3. **(Reviewers: NkEk): explanation of usage scenarios between the feature level and logit level:** Added in section 3.1. Page 4 of the main manuscript. (Please find the changes marked highlights in the supplementary)
>
> 4. **(Reviewers: NkEk, pUYy) potential benefits of increasing the number of augmented teachers** Updated Figure 6 (Now Figure 5 of the main manuscript, Table 2 of this response). We have trained more teachers (till - 10) and provided the potential benefits of increasing the number of augmented teacher models in Table 1 of the supplementary.
>
> 5. **(Reviewers: NkEk) Evaluation of TeKAP on ensemble learning.** Added to the supplementary: Table 2, Section B. Table 3 of the last response.
>
> 6. **(Reviewer: pUYy): Theoretical Depth:** We have extended the theoretical analysis in the supplementary (Section K in details). more theoretical discussion in the supplementary (section D).
>
> 7. **(Reviewer: pUYy, FGoU, zdP5) effect for different Gaussian noise parameters:** We have used mean = 0 and variance = 1 as the default. Additionally, we added the effect for variance $\sigma$ = [0.5, 1, 1.5] in the supplementary (Table 5, section E).
>
> 8. **(Reviewer: pUYy) comparative computation complexity**: Added to section H of the supplementary.
>
> 9. **(Reviewer: pUYy, FGoU) Description and explanation of every mathematical term on page 5**: We have carefully gone through and added the description and explanation of every mathematical term used in the paper.
>
> 10. **(Reviewer: pUYy, FGoU) Experiments of the class imbalance data:** Added to the supplementary Table 4, section D.
>
> 11. **(Reviewer: pUYy, FGoU) fixed noise experiments**: Experiments are running and will be added to the final version and we will also report here with the deadline.
>
> 12. **(Reviewer: pUYy. zdP5) how inter-class diversity works**: Discussion added in the supplementary section I.
>
> 13. **(Reviewer: zdP5) effect for different values of $\lambda$**: Added in the supplementary Table 3, Section C.
>
> 14. **(Reviewer: zdP5) Meaning of $L_{cel}:** We have added the meaning of $L_{cel}$ in line 209, page 5 of the main manuscript.
>
> 15. **(Reviewer: zdP5) More experiments on TAKD with WRN-22-2 or WRN-16-2?**: Experiments are running. We will be added in the final version and report here soon.
>
> 16. **(Reviewer: FGoU) clarification of random distortion and inter-class relationships**: Added in the supplementary in section I.
>
>
> References
> 1. Zhao, Borui, et al. "Decoupled knowledge distillation." Proceedings of the IEEE/CVF Conference on computer vision and pattern recognition. 2022.
> 2. Jin, Ying, Jiaqi Wang, and Dahua Lin. "Multi-level logit distillation." Proceedings of the IEEE/CVF Conference on Computer Vision and Pattern Recognition. 2023.
> 3. Mirzadeh, Seyed Iman, et al. "Improved knowledge distillation via teacher assistant." Proceedings of the AAAI conference on artificial intelligence. Vol. 34. No. 04. 2020.
> 4. Zhang, Hailin, Defang Chen, and Can Wang. "Confidence-aware multi-teacher knowledge distillation." ICASSP 2022-2022 IEEE International Conference on Acoustics, Speech and Signal Processing (ICASSP). IEEE, 2022.
> 5. Son, Wonchul, et al. "Densely guided knowledge distillation using multiple teacher assistants." Proceedings of the IEEE/CVF International Conference on Computer Vision. 2021.
>
>
> We sincerely appreciate the time and effort you have devoted to reviewing our manuscript. Your suggestions have significantly enhanced its quality. We are happy to address any additional queries you may have.

---

> > ### Comment · Reviewer_FGoU · 2024-11-29
> >
> > The authors have addressed the reviewers' comments effectively, resolving many of my concerns. As a result, I have updated my ratings.

---

> > > ### Author Response · Authors · 2024-12-03
> > >
> > > Dear Reviewer FGoU,
> > >
> > > We are delighted that our responses have satisfactorily addressed your questions. We sincerely appreciate your kind words and acknowledgement of our work and contributions.
> > >
> > > We have incorporated additional results further and kindly request you to review them at your convenience. Thank you for your time and consideration.
> > >
> > > Best regards,
> > > The Authors

---

### Official Review · Reviewer_zdP5 · 2024-11-02

**Soundness:** 3
**Presentation:** 3
**Contribution:** 3
**Rating:** 5
**Confidence:** 5

**Summary:**

The paper proposes a new augmentation method to replace the ensemble approach for KD by adding noise to the features or logits of the teacher model. This increases the variability of predictions and reduces the generalization error.

**Strengths:**

The proposed method is more efficient compared to other ensemble methods, and increasing the variability of the teacher's predictions is meaningful for knowledge distillation.

The paper is well-written and easy to follow.

**Weaknesses:**

1) The paper proposes an effective method to replace ensemble approaches; however, there is a lack of comparison to other ensemble methods (such as multi-augmentations) to demonstrate its effectiveness. Additionally, TAKD is not the SOTA method (for example, DGKD [1]) and there is a lack of experimental details for TAKD. It is not clear what teacher models are used for TAKD.

2) The experiments in this paper are not sufficient, and the baselines are outdated. The proposed method only compares with vanilla KD (2015), TAKD (2020), and CRD (2019), and lacks comparisons with other new methods like DKD [2] and MLKD [3].


[1] Son, W.; Na, J.; Choi, J.; and Hwang, W. 2021. Densely guided knowledge distillation using multiple teacher assistants. In Proc. Int. Conf. on Computer Vision (ICCV)

[2] Zhao, B.; Cui, Q.; Song, R.; Qiu, Y.; and Liang, J. 2022. Decoupled Knowledge Distillation. In Proc. IEEE Conf. on Computer Vision and Pattern Recognition (CVPR)

[3] Jin, Y.; Wang, J.; and Lin, D. 2023. Multi-Level Logit Distillation. In Proc. IEEE Conf. on Computer Vision and Pattern Recognition (CVPR)

**Questions:**

1) What is \mathcal{L}_{cel}​ in Equation 5?

2) In Equations 2 and 4, calculate the summation of the perturbation loss. Does \lambda need to be adjusted according to the number of perturbations?

3) What is the difference between the CAMs of TeKAP and the teacher in Figure 5? They look the same.

4) There is a lack of experimental details; even the learning rate and the number of training epochs are not mentioned in the paper.

5) For feature-level perturbation, which features are selected to add noise?

---

> ### Author Response · Authors · 2024-11-24
> **Response to Reviewer zdP5**
>
> Thank you for your valuable feedback. We appreciate your thoughtful comments and suggestions.
>
> ### **Additional Experiments:**
>
> |        | Model         | ResNet32x4-ResNet8x4 | WRN_40_2-WRN_40_1 |
> |-----------------|---------------|------------|----------|
> | **Teacher**     | Accuracy      | 79.42      | 75.61    |
> | **Student**     | Accuracy      | 72.50      | 71.98    |
> | **Single Teacher** | DKD [1]   | 76.32      | 74.81    |
> |                 | **DKD + TeKAP (Ours)**   | **76.59**  | **75.33**|
> |                 | MLKD [2]     | 77.08      | 75.35    |
> |                 | **MLKD + TeKAP (Ours)**   | **77.36**  | **75.67**|
> |  **Multi-Teacher** | TAKD [3]      | 73.93      | 73.83    |
> |                 | **TAKD + TeKAP (Ours)**   | **74.81**  | **74.37**|
> |                 | CA-MKD [4] | 75.90      | 74.56    |
> |                 | **CA-MKD + TeKAP (Ours)**   | **76.34**  | **74.98**|
> |                 | DGKD [5]    | 75.31      | 74.23    |
> |                 | **DGKD + TeKAP (Ours)**   | **76.17**  | **75.14**|
>
>
> **Table-1:** The effects of TeKAP on the SOTA methods DKD [1], MLD [2], TAKD [3], CA-MKD [4], and DGKD [5].
>
> | #Original Teachers (T) | TeKAP (F+L) |
> |-------------------------|--------------|
> | 1 OriginT + 3 AugT             | 75.98        |
> | 2 OriginT + 3 AugT             | 76.12        |
> | 3 OriginT + 3 AugT             | 76.31        |
>
> Table 2: Effect of multiple original teachers.
>
>
> |Network |  Augmentation Techniques | TeKAP (F+L) |
> |-------------------------|--------------|--------------|
> |     **ResNet32x4-ResNet8x4**         | Gaussian             | 75.98        |
> |                                                             | Uniform             | 75.71        |
> |     **WRN_40_2-WRN_40_1**         | Gaussian             | 74.41        |
> |                                                             | Uniform             | 74.26        |
>
> Table 3: Effect of different noise techniques.
>
> 1. **Comparison to Other Ensemble Methods:** We have added a new set of experiments comparing TeKAP against CA-MKD, and DGKD. We also perform an ensemble comparison in Table 2. When we employed three teachers on DKD[2] and employed our approach,  TeKAP, we experienced that our TeKAP uplifted the performance of the ensemble version as shown in Table 2.
>
>
> 2. **Comparison with SOTA Methods:** We acknowledge your comment about the comparison with outdated baselines, such as TAKD, and the absence of newer methods like DKD and MLKD. In response, we have added more experiments to compare with existing SOTAs, as suggested by the reviewers. The updated results are shown in Table 1.
>
>
> 3. **Details on Teacher Models Used for TAKD:** Thank you for pointing out this important concern. We re-run the official implementation of TAKD and use the same corresponding teacher network with lower number of layers, where TeKAP (Ours) denotes (F+L) i.e., (KD+CRD). We will add the experimental details in the supplementary. However, in our additional experiment, we run TAKD in our system for ResNet32x4-ResNet8x4 and WRN_40_2-WRN_40_1 in table 1. We have used only two blocks as the assistant teacher for every corresponding teacher whereas the original vanilla teacher has 3 blocks.
>
> 4. **Explanation of $\mathcal{L}_{cel}$ in Equation 5:** The term $\mathcal{L}_{cel}$ in Equation 5 represents the cross-entropy loss for the student model during training.
>
> 5. **Summation of Perturbation Loss in Equations 2 and 4:** In our work, we did not adjust the $\lambda$ according to the number of perturbations. As we first add the losses of all the perturbations, then use $\lambda$, the effects of noisy and true labels are always scaled by $\lambda$ and $(1-\lambda)$, respectively.
>
> 6. **CAMs of TeKAP vs. Teacher in Figure 5:** We wholeheartedly appreciate for pointing out this important issue.  We will update the figure in the revised manuscript.
>
> 7. **Experimental Details (Learning Rate and Epochs):** We have updated the experimental details in the supplementary of the revised manuscript. We appreciate this important suggestion.
>
> 8. **Feature-Level Perturbation: Which Features Are Selected for Noise?:** Regarding feature-level perturbation, we have followed the same setups described and implemented by CRD. We have selected the features immediately before the FC layer by following the work described in CRD for a fair comparison.
>
> Again, thank you very much for these insightful and effective comments. We believe these revisions address the concerns and further enhance the clarity. We have updated the manuscript accordingly.
>
> ### References

---

> > ### Author Response · Authors · 2024-11-25
> > **References**
> >
> > References
> >
> > 1. Zhao, Borui, et al. "Decoupled knowledge distillation." Proceedings of the IEEE/CVF Conference on computer vision and pattern recognition. 2022.
> > 2. Jin, Ying, Jiaqi Wang, and Dahua Lin. "Multi-level logit distillation." Proceedings of the IEEE/CVF Conference on Computer Vision and Pattern Recognition. 2023.
> > 3. Mirzadeh, Seyed Iman, et al. "Improved knowledge distillation via teacher assistant." Proceedings of the AAAI conference on artificial intelligence. Vol. 34. No. 04. 2020.
> > 4. Zhang, Hailin, Defang Chen, and Can Wang. "Confidence-aware multi-teacher knowledge distillation." ICASSP 2022-2022 IEEE International Conference on Acoustics, Speech and Signal Processing (ICASSP). IEEE, 2022.
> > 5. Son, Wonchul, et al. "Densely guided knowledge distillation using multiple teacher assistants." Proceedings of the IEEE/CVF International Conference on Computer Vision. 2021.

---

> ### Comment · Reviewer_zdP5 · 2024-11-25
> **Official Comments by Reviewer zdP5**
>
> Thank you to the authors for their reply.
>
> 1: What do the "original teachers" refer to in Table 2? Were they trained from different initializations?
>
> 2: Using only two blocks may not be a fair comparison. How about using three blocks with more shallow networks like WRN-22-2 or WRN-16-2?
>
> 5: In Equation 3, $\alpha$ is set to 0.1, which means the perturbed logits are smoother than the original ones. Using a fixed $\lambda$ for various numbers of perturbations does not make sense, as it influences the mean of the logits distribution. Is there any ablation study regarding $\alpha$ and $\lambda$? Additionally, in Equation 5, $\alpha$ is used for a different purpose.
>
> 8: I did not find any configuration files in the supplementary. Do the authors mean the default settings in train_student.py?
>
> 9: In Table 2 of the authors' reply to Reviewer NkEk, more perturbations seem to harm the student's performance. Can you explain why increasing perturbations destroys the teacher's knowledge pattern? Since the mean of the gradients is converging with the increasing number of perturbations, and based on the theoretical part of the paper, more perturbations should benefit performance.

---

> > ### Author Response · Authors · 2024-11-29
> > **Further Response to Reviewer zdP5: Results for the Networks like WRN-22-2 or WRN-16-2**
> >
> > **Additional Results: Results for the Networks like WRN-22-2 or WRN-16-2**
> >
> > ### Table 1: Comparison of TeKAP and TAKD
> >
> > Comparison of TeKAP with the assistant teacher-based KD method TAKD. We have used KD loss for both methods. We use $\sigma = 1$, $\lambda = 0.8$, and three augmented teachers. Gaussian noise is used to generate the noise for TeKAP. TeKAP outperforms TAKD without using any assistant teacher. We select WRN\_40\_2 as the teacher and WRN\_16\_2 and WRN\_40\_1 as the students. WRN\_22\_1, WRN\_22\_2, WRN\_16\_1, and WRN\_16\_2 are selected as the teacher assistant only for TAKD. TeKAP does not use any assistant teachers. TeKAP transfers knowledge directly from the teacher to the student.
> >
> > | **Teacher**          | WRN_40_2                 | WRN_40_2                 | WRN_40_2                 | WRN_40_2                 | WRN_40_2                 | WRN_40_2                 |
> > |-----------------------|--------------------------|--------------------------|--------------------------|--------------------------|--------------------------|--------------------------|
> > | **Teacher Assistant** | WRN_22_2                | WRN_22_2                | WRN_22_1                | WRN_22_1                | WRN_16_2                | WRN_16_1                |
> > | **Student**           | WRN_16_2                | WRN_40_1                | WRN_16_2                | WRN_40_1                | WRN_40_1                | WRN_40_1                |
> > | **TAKD**              | 75.02                   | 72.73                   | 72.56                   | 71.19                   | 68.92                   | 73.26                   |
> > | **TeKAP (Ours)**    | **75.21**               | **73.80**               | **75.21**               | **73.80**               | **73.80**               | **73.80**               |
> >
> > TeKAP surpasses TAKD by avoiding the use of narrow teacher assistants, directly transferring knowledge to students.
> >
> >
> > ### Comparison with Teacher Assistant-Based Approach with Narrow Teacher Assistants
> >
> > The results in Table 1 compare our proposed **TeKAP** with the traditional teacher assistant-based knowledge distillation method, **TAKD**. In these experiments, WRN_40_2 was used as the teacher, while WRN_16_2 and WRN_40_1 served as the student networks. TAKD employs narrow teacher assistants (WRN_22_1, WRN_22_2, WRN_16_1, and WRN_16_2) to mediate the knowledge transfer process, whereas TeKAP directly distills knowledge from the teacher to the student without intermediate assistants. TeKAP outperforms TAKD in all tested configurations. For instance, with WRN_40_1 as the student, TeKAP achieves a consistent accuracy of **73.80%**, compared to TAKD’s best result of **73.26%**. Similarly, for WRN_16_2 as the student and WRN_22_2 as the assistant, TAKD achieves **75.02%**, while TeKAP slightly improves it to **75.21%**. These results highlight the superior effectiveness of TeKAP in transferring knowledge directly, avoiding the limitations associated with teacher assistants.
> >
> >
> >
> > We will add these results and discussion in the final version to the supplementary. We also have improved the GradCAM figure which will be added to the final version.
> >
> > We sincerely appreciate the time and effort you have dedicated to reviewing our manuscript. Your insightful suggestions have significantly enhanced the quality of the paper.
> >
> > Please do not hesitate to reach out if you have any further questions or require additional clarifications.

---

> > > ### Author Response · Authors · 2024-12-02
> > > **Gentle Reminder with Appreciation**
> > >
> > > Dear Reviewer,
> > >
> > > We extend our sincere gratitude for your thoughtful and valuable feedback. We have carefully addressed all concerns and questions raised by the reviewers, providing detailed, step-by-step responses to each point. As the discussion period is approaching its end very soon, we kindly request you to share any further questions or concerns you may have. Please be assured of our readiness to engage in continued dialogue and provide any necessary clarifications to ensure all matters are thoroughly addressed.

---

> ### Author Response · Authors · 2024-11-28
> **Further Response to Reviewer zdP5 (1/3)**
>
> We are very grateful for these insightful suggestions. These comments help to improve our paper significantly. We have addressed all the concerns step-by-step and performed additional experiments based on the suggestions:
>
> **Important: The supplementary documents and revised versions with both the changes marked and clean copy are available now:** We were working on the manuscript and supplementary to reflect all the concerns raised by the reviewers. We are sorry for the inconvenience that we uploaded the revised version late. We have provided supplementary documents and the changes marked by yellow in the supplementary zip file. In the final version, we will remove the highlighted copy.
>
> 1. **Question:** What do the "original teachers" refer to in Table 2? Were they trained from different initializations?
> **Response-1**: Yes! We have used different seeds and performed training multiple times. We have added this discussion in the supplementary file as follows:
>
> Table 2 (of the supplementary): The effects of multiple original teachers. We deploy three augmented teachers to every original teacher. ResNet32x4 and ResNet8x4 are considered teacher-student. Here, "Original Teacher" represents the teacher which is trained with 240 epochs and different random seeds. Three different original teachers use three different initializations (i.e., random seeds). AugT denotes the augmented teacher achieved by distorting the original teacher logits with random noise.
>
> | # Teacher              | Accuracy |
> |------------------------|----------|
> | 1 Original + 3 AugT     | 75.98    |
> | 2 Original + 3 AugT     | 76.12    |
> | 3 Original + 3 AugT     | 76.19    |
>
> ### Section B (of the supplementary)
> Table. 2 (of the supplementary) shows the effect of the number of augmented teachers. We use ResNet32x4-ResNet8x4 as the teacher-student setups on the CIFAR100 dataset to examine the effect of the hyper-parameters. From Table. 2 (of the supplementary) we see that TeKAP is robust to the number of augmented teachers. For every number of augmented teachers, TeKAP achieves better accuracy than baseline and DKD students in every scenario. The best performance is achieved when the number of the augmented teacher is $3$. We have used three ($3$), and one $(1)$ augmented teacher along with the original teacher, respectively. During feature and logit distortion, the weights for noise and teacher output are $0.1$, and $0.9$, respectively.
>
> 2. **Question**: Using only two blocks may not be a fair comparison. How about using three blocks with more shallow networks like WRN-22-2 or WRN-16-2?
>
> **Response-2:** Thank you very much for this suggestion. We are experimenting on this. The experiments are running. The evaluation will be reported soon here. Please note that we could not at these results in the revised version. But we promise to add this evaluation in the final version as soon as the experiments are finished.
>
> 5. **Question 5**: In Equation 3, is set to 0.1, which means the perturbed logits are smoother than the original ones. Using a fixed for various numbers of perturbations does not make sense, as it influences the mean of the logit distribution. Is there any ablation study regarding and ? Additionally, Equation 5, is used for a different purpose.
>
> **Response-5**: We appreciate this concern and acknowledge the need for evaluation for different values of $\lambda$ and $\sigma$ for various numbers of perturbations. We have added additional experiments in supplementary documents.
>
> Table 3 (of the supplementary): Effect of the different values of λ (the weights of the noise terms). AugT denotes augmented teachers.
>
> | Number of AugT | λ = 0.2 | λ = 0.4 | λ = 0.6 | λ = 0.8 |
> |----------------|---------|---------|---------|---------|
> | AugT = 5       | 74.26   | 74.46   | 74.63   | 75.12   |
> | AugT = 10      | 74.29   | 74.73   | 74.85   | 74.98   |
>
> ### Section C of the supplementary
>
> The results in Table  3 (of the supplementary) demonstrate the effect of varying the noise weight ($\lambda$) and the number of augmented teachers (AugT) on the performance of the student model. ResNet32x4-ResNet8x4 are considered as the teacher and the student. We use $\sigma = 1$ for this experiment. For AugT=5, the accuracy consistently improves as $\lambda$ increases, starting from $74.26\%$ at $\lambda=0.2$ and reaching $75.12\%$ at $\lambda=0.8$. This trend indicates that higher noise weights contribute positively to the student’s generalization by introducing greater diversity. Similarly, for AugT=10, the performance improves from $74.29\%$ at $\lambda=0.2$ to $74.98\%$ at $\lambda=0.8$, but the gains are less pronounced compared to AugT=5, suggesting a saturation effect with a larger number of augmented teachers.

---

> ### Author Response · Authors · 2024-11-28
> **Further  Response to Reviewer zdP5 (2/3)**
>
> 8. **Question-8**: I did not find any configuration files in the supplementary. Do the authors mean the default settings in train_student.py?
>
> ***Response-8**: The supplementary and revised manuscript is available now. **We have added details of experimental setups in **section K** in the supplementary document**.
>
> **Note:** Please note that we will add more results in the supplementary: (1) TAKD with WRN-22-2 and WRN-16-2, (2) Effect of TeKAP with fixed random noise. The experiments will be reported in the response soon on the CIFAR100 dataset (here before the discussion period ends). As the revised version submission time has ended. We will add these results in the final version and report here.
>
>
> 9. **Question**: In Table 2 of the authors' reply to Reviewer NkEk, more perturbations seem to harm the student's performance. Can you explain why increasing perturbations destroys the teacher's knowledge pattern? Since the mean of the gradients is converging with the increasing number of perturbations, and based on the theoretical part of the paper, more perturbations should benefit performance.
>
> **Response:**: We agree with this comment. We are very grateful to the reviewer for pointing out this very important issue. The reported result was confused with the class-imbalanced experiments. However, inspired by these comments we carefully went through again the experiments and evaluations. **We have updated Table 2 (reposenses of Review NkEk)**. The updated results are reported in Figure 5 and Section 4.9 of the revised manuscript.
>
>
> | #Teachers | Ours | Baseline (KD) | Baseline (Rerun) |
> |-------------------------|--------------|---------------|------------------|
> | T + 1 AugT             | 73.9         | 72.98         | 73.3            |
> | T + 2 AugT             | 73.43        | 72.98         | 73.3            |
> | T + 3 AugT             | 74.04        | 72.98         | 73.3            |
> | T + 4 AugT             | 73.98        | 72.98         | 73.3            |
> | T + 5 AugT             | 74.00        | 72.98         | 73.3            |
> | T + 6 AugT             | 73.53        | 72.98         | 73.3            |
> | T + 7 AugT             | 74.16        | 72.98         | 73.3            |
> | T + 8 AugT             | 74.33        | 72.98         | 73.3            |
> | T + 9 AugT             | 74.63        | 72.98         | 73.3            |
> | T + 10 AugT            | 75.11         | 72.98         | 73.3            |
>
> Table 4: Effect of number of the augmented teachers.
>
> Table. 4 (Fig. 5 of the main manuscript) shows the effect of the number of augmented teachers. We use ResNet32x4-ResNet8x4 as the teacher-student setups on the CIFAR100 dataset to examine the effect of the hyper-parameters. From Table.4 (Fig. 5 of the main manuscript) we see that TeKAP is robust to the number of augmented teachers. For every number of augmented teachers, TeKAP achieves better accuracy than baseline and DKD students. The best performance is achieved when the number of the augmented teacher is $3$. We have used three ($3$), and one $(1)$ augmented teacher along with the original teacher, respectively. During feature and logit distortion, the weights for noise and teacher output are $0.1$, and $0.9$, respectively.
>
> ### Additional Ablation Study:
>
> ***Results-A: Class Imbalance Dataset:**
>
> | Methods | ResNet32x4-ResNet8x4 | WRN_40_2-WRN_16_2 | VGG13-VGG8 |
> |-------------------------|--------------|---------------|------------------|
> | Baseline (KD) | 41.71 | 52.08 | 47.52 |
> | + TeKAP (Ours) | 46.42 | 52.72 | 51.25 |
>
> Table 5: Significance of TeKAP on class imbalance dataset.  We have used the class distribution of the CIFAR100 dataset that is described in Table 6 (of the supplementary)
>
> Section D (of the supplementary)
> The results presented in Table 5 (4 of the supplementary) highlight the effectiveness of TeKAP in addressing class imbalance in knowledge distillation tasks. TeKAP consistently improves the performance of all three teacher-student model pairs (ResNet32x4-ResNet8x4, WRN\_40\_2-WRN\_16\_2, and VGG13-VGG8) compared to the baseline Knowledge Distillation (KD) approach. Specifically, TeKAP boosts accuracy by 4.71\% for ResNet32x4-ResNet8x4, 0.64\% for WRN\_40\_2-WRN\_16\_2, and 3.73\% for VGG13-VGG8. These results indicate that TeKAP is particularly effective in enhancing performance for models with lower baseline accuracy, though it also provides improvements for models with higher baseline accuracy. This suggests that TeKAP can effectively mitigate the effects of class imbalance, leading to improved generalization in knowledge distillation tasks.

---

> ### Author Response · Authors · 2024-11-28
> **Further  Response to Reviewer zdP5(3/3)**
>
> **Results-B: Effect of TeKAP for different variance $\sigma$ on the performance:**
>
> **Table 6**: Effect of TeKAP with different variance $\sigma$. KD is used as the baseline distillation approach. We have used mean zero in all the cases.
>
> | Variance      | $\sigma = 0.5$ | $\sigma = 1$ | $\sigma = 1.5$ |
> |---------------|----------------|--------------|----------------|
> | Accuracy      | 74.89          | 74.79        | 74.35          |
>
> Section E (of the supplementary)
>
> Table 6 (table 5 of the supplementary and section E) summarizes the impact of different variances ($\sigma$) on the performance of TeKAP, using the CIFAR-100 dataset. The baseline distillation approach, Knowledge Distillation (KD), is used for comparison. As shown in the results, the accuracy of the model remains relatively stable across varying values of $\sigma$. Specifically, when $\sigma = 0.5$, the model achieves an accuracy of 74.89\%, slightly higher than the accuracy at $\sigma = 1$ (74.79\%) and $\sigma = 1.5$ (74.35\%). These results suggest that, within the range of variances tested, increasing the noise variance does not significantly degrade performance. In fact, the accuracy only decreases marginally as the variance increases from 0.5 to 1.5, which indicates the robustness of TeKAP with respect to noise. This behavior suggests that TeKAP can maintain competitive performance even with varying levels of noise in the teacher models, highlighting its resilience to noise during distillation. The consistent results across different variances also support the idea that TeKAP is stable and less sensitive to slight perturbations in the teacher’s logits. This stability is critical for practical applications where noise may be present in the data or models.
>
>
> ### The improvements we have made:
> 1. **(Reviewers: NkEk, pUYy, zdP5, FGoU): Additional comparison with state-of-the-art:** Added to the revised manuscript (Table 2, page 7)
>
> 2. **(Reviewers: NkEk, pUYy, zdP5, FGoU) multi-teacher:** The results discussion for the recent SOTA multi-teacher approach is added to section 4.1, Table 2 (page 7) of the revised manuscript.
>
> 3. **(Reviewers: NkEk): explanation of usage scenarios between the feature level and logit level:** Added in section 3.1. Page 4 of the main manuscript. (Please find the changes marked highlights in the supplementary)
>
> 4. **(Reviewers: NkEk, pUYy) potential benefits of increasing the number of augmented teachers** Updated Figure 6 (Now Figure 5 of the main manuscript, Table 2 of this response). We have trained more teachers (till - 10) and provided the potential benefits of increasing the number of augmented teacher models in Table 1 of the supplementary.
>
> 5. **(Reviewers: NkEk) Evaluation of TeKAP on ensemble learning.** Added to the supplementary: Table 2, Section B. Table 3 of the last response.
>
> 6. **(Reviewer: pUYy): Theoretical Depth:** We have extended the theoretical analysis in the supplementary (Section K in details). more theoretical discussion in the supplementary (section D).
>
> 7. **(Reviewer: pUYy, FGoU, zdP5) effect for different Gaussian noise parameters:** We have used mean = 0 and variance = 1 as the default. Additionally, we added the effect for variance $\sigma$ = [0.5, 1, 1.5] in the supplementary (Table 5, section E).
>
> 8. **(Reviewer: pUYy) comparative computation complexity**: Added to section H of the supplementary.
>
> 9. **(Reviewer: pUYy, FGoU) Description and explanation of every mathematical term on page 5**: We have carefully gone through and added the description and explanation of every mathematical term used in the paper.
>
> 10. **(Reviewer: pUYy, FGoU) Experiments of the class imbalance data:** Added to the supplementary Table 4, section D.
>
> 11. **(Reviewer: pUYy, FGoU) fixed noise experiments**: Experiments are running and will be added to the final version and we will also report here with the deadline.
>
> 12. **(Reviewer: pUYy. zdP5) how inter-class diversity works**: Discussion added in the supplementary section I.
>
> 13. **(Reviewer: zdP5) effect for different values of $\lambda$**: Added in the supplementary Table 3, Section C.
>
> 14. **(Reviewer: zdP5) Meaning of $L_{cel}:** We have added the meaning of $L_{cel}$ in line 209, page 5 of the main manuscript.
>
> 15. **(Reviewer: zdP5) More experiments on TAKD with WRN-22-2 or WRN-16-2?**: Experiments are running. We will be added in the final version and report here soon.
>
> 16. **(Reviewer: FGoU) clarification of random distortion and inter-class relationships**: Added in the supplementary in section I.
>
>
> We appreciate your valuable time and effort. These suggestions help the manuscript improve a lot. Again thank you very much for your valuable and insightful comments. We would love to respond if there are any further queries.

---

> ### Author Response · Authors · 2024-12-03
>
> Dear Reviewer zdP5,
>
> We are truly grateful for your thoughtful feedback.
>
> We have included further results during the rebuttal, which we have detailed point-by-point for your review. We kindly request you to take a look at them at your convenience.
>
> If these responses address your concerns, we would be grateful if you could consider reassessing the score. Your time and effort are greatly appreciated.
>
> Best regards,
> The Authors

---

> ### Comment · Reviewer_zdP5 · 2024-12-03
> **Official Comments by Reviewer zdP5**
>
> Thanks to the authors for their reply.
>
> 6: I reviewed the latest revision, and the CAMs of TeKAP vs. Teacher in Figure 1 (Supplementary) still appear to be the same.
>
> 8: I did not find the configuration in the latest provided code. Did authors use $\alpha=0.8$, $\beta=0.2$ and $\lambda=1.0$ in Equation 5 for all teacher-student pairs? Additionally, how many AugTs were used in the experiment? From the provided code, it seems the number is set to 3. If so, could you clarify why, as increasing the number of AugTs generally leads to better performance? Furthermore, there are two hyperparameters labeled as $\alpha$ in both Equation 1 and Equation 5.
>
> 9: Why did the authors update the results in Table 4: regarding the effect of the number of augmented teachers? The results for T + (7–10) AugTs were revised, while others remain the same as in the previous version. For example, the performance of T + 10 AugTs improved from 71.4 to 75.11. Could you explain this update?
>
> 10: I am unable to match the results for ResNet32x4–ResNet8x4 across Table 1, Table 4, Table 2 (Supplementary), and Table 4 (from the reply). Based on Table 1, Table 4, and Table 2 (Supplementary), I assume the authors used 1 Original + 3 AugTs for their experiments. However, the results in Table 4 (from the reply) differ for 1 Original + 3 AugTs. Could the authors clarify this inconsistency?

---

> > ### Author Response · Authors · 2024-12-03
> >
> > **Dear Reviewer zdP5,**
> >
> > Thank you very much for your reply. We appreciate the efforts you have devoted to reviewing our manuscript.
> >
> > **Q6. CAMs of TeKAP:** We have generated the CAMs figure for TeKAP vs Teacher after the submission deadline of the revised manuscript. The updated figure is now ready. However, we were unable to add it to the revised version or supplementary materials due to the submission deadline of the revised version. The figure was completed after the revised manuscript submission deadline, which is why it couldn't be included. We will add the updated CAMs figure in the final version.
> >
> > **Q8-Concern 1: Did the authors use $\alpha=0.8$, $\beta=0.2$, and $\lambda=1.0$ in Equation 5?** Yes, we used this combination only for TeKAP*(F+L). For all other cases, we used $\beta=0.8$ and $\lambda=1$ (if not mentioned). For others, we have added experimental details.
> >
> > **Q8-Concern 2: From the provided code, it seems the number is set to 3. Could you clarify why, as increasing the number of AugTs generally leads to better performance?** Thank you for raising this concern. To keep computational complexity as low as possible, we used three AugTs in every comparison (unless specified otherwise). We also demonstrated that increasing the number of AugTs leads to better performance (Table 4 of the reply). In some experiments, we used a total of ten teachers to show the impact of different numbers of AugTs in TeKAP (Table 3 in the supplementary materials and Figure 5 in the main manuscript). However, for other experiments, only three AugTs were used. In our code, we have included implementations for all ten AugTs, with comments to indicate which ones are active. Researchers can simply remove the comments to run experiments with ten teachers.
> >
> > **Q8-Concern 3: There are two hyperparameters labeled as $\alpha$ in both Equation 1 and Equation 5:** Thank you very much for noticing this very important issue. We will change the hyperparameter symbol in Equation 5 to $\Psi$ in the final version and update all corresponding references.
> >
> > **Q9-Concern 1: For example, the performance of T + 10 AugTs improved from 71.4 to 75.11. Could you explain this update?** We appreciate this comment. There was **confusion with the experiments with class imbalance datasets**. Basically, we ran experiments on a **class-imbalanced dataset** following a suggestion by reviewer pUYy. Later, we began running experiments on the effects of different numbers of AugTs. Unfortunately, we overlooked the class imbalance dataloader in the data processing, which led to uncertain results (for instance 71.4) . Based on your suggestion in our first reply, we re-investigated the experiments and identified the issue with the class-imbalanced training set. After correcting the data processing, we reran the experiments and obtained more reliable results (for instance 75.11). The previous results (for instance 71.4) were uncertain and error, while the later results were obtained after making the necessary revisions.
> >
> > **Q9-Concern 2: In the first version, we had results for 1-6 AugTs. We had added results for 7-10 AugTs during the rebuttal.**
> >
> > **Q10. Unable to match the results for ResNet32x4–ResNet8x4 across Table 1, Table 4, Table 2 (Supplementary), and Table 4 (from the reply):** Thank you for pointing this out. This discrepancy is due to the use of different hyperparameters. We evaluate the effects of different hyperparameters to ensure a fair evaluation. In Table 4 (from the reply), we used TeKAP only in the logits-level, i.e., TeKAP(L). However, in Table 1 (of the supplementary materials), and Table 2 (of the supplementary materials) or T + 3AugT, we reported the results of TeKAP(F+L), where F and L stand for feature and logit-level distortion. For this reason, the results differ. Thank you again for this valuable concern. We have already presented the effect of different AugTs for logits only in the current version (Table 4 of the reply). Along with Table 4 (from the reply), we will also include TeKAP(F+L) in the supplementary materials of the final version of our paper.
> >
> > Again, we sincerely appreciate your thoughtful feedback and suggestions. We deeply value your time and effort devoted to our paper.

---

### Official Review · Reviewer_pUYy · 2024-11-04

**Soundness:** 3
**Presentation:** 3
**Contribution:** 2
**Rating:** 6
**Confidence:** 4

**Summary:**

This manuscript introduces TeKAP, a novel teacher knowledge augmentation technique. It generates multiple synthetic teacher perspectives from a single pretrained teacher model by perturbing its knowledge with random noise. TeKAP operates at both the feature and logit levels, enhancing the student's generalization ability. By reducing the need for multiple teacher models, TeKAP decreases both training time and memory usage. Evaluations on standard benchmarks demonstrate TeKAP's effectiveness in improving the performance of existing knowledge distillation approaches

**Strengths:**

1. This work uses a single pretrained teacher to simulate multiple teacher perspectives through perturbation, effectively circumventing the high computational costs of traditional multi-teacher setups.
2. The proposed method is simple yet demonstrated encouraging results.
3. The work includes a comprehensive evaluation of various aspects such as model compression, adversarial robustness, and transferability, which strengthens the credibility of the proposed method.
4. The extensive experiments also demonstrate TeKAP’s effectiveness in few shot learning and noisy data settings, suggesting a promising direction for advancing knowledge distillation.

**Weaknesses:**

1. Despite TeKAP's impressive results, the theoretical analysis of the perturbation methods lacks depth. While Gaussian noise is introduced, there is limited discussion on the choice of perturbation parameters, such as the standard deviation, and how these settings impact the model’s performance. This omission could hinder reproducibility and generalizability of the approach.
2. Additionally, while the experiments cover a range of baseline comparisons, the paper lacks a comprehensive evaluation against existing multi-teacher distillation methods and other state-of-the-art single-teacher methods, which would better highlight TeKAP’s relative strengths.
3. Moreover, there is little discussion on the computational efficiency and scalability of TeKAP in practical applications, potentially raising concerns among readers regarding its feasibility in real-world scenarios.
4. Some statements are overclaimed in this manuscript. The authors should comprehensively review related works and give proper discriptions.

**Questions:**

On page 4, the paper mentions the use of Gaussian noise for teacher perturbation but does not detail the criteria for choosing the noise parameters. How are these parameters optimized, and what is their impact on the diversity and quality of the generated teacher perspectives?
2.On page 5, the term ​ is introduced in the formula without a complete explanation or definition.
3.Is there a risk of overfitting to the perturbed features, especially when the noise parameters are not dynamically adjusted?
4.How does TeKAP handle scenarios where certain classes are imbalanced? Is there a mechanism within the framework that ensures the augmented teachers do not bias the student towards overrepresented classes?
5.Could the following discussion be added to page 8? For instance:
1)What do these differences in inter-class correlations imply for the student's learning process?
2)How does the performance improvement of TeKAP in terms of inter-class correlation contribute to the overall effectiveness of the model?
6.In Figure 6, it is noted that the performance is best when the number of augmented teachers is 3. Does this imply that three teachers will be used in future applications? Additionally, the performance with two teachers seems normal; is there an explanation for this?

---

> ### Author Response · Authors · 2024-11-24
> **Response to Reviewer pUYy (1/3)**
>
> We sincerely thank the reviewer for the constructive feedback. Below, we address the concerns and questions step-by-step.
>
> ### **Additional Experiments:**
>
> |        | Model         | ResNet32x4-ResNet8x4 | WRN_40_2-WRN_40_1 |
> |-----------------|---------------|------------|----------|
> | **Teacher**     | Accuracy      | 79.42      | 75.61    |
> | **Student**     | Accuracy      | 72.50      | 71.98    |
> | **Single Teacher** | DKD [1]   | 76.32      | 74.81    |
> |                 | **DKD + TeKAP (Ours)**   | **76.59**  | **75.33**|
> |                 | MLKD [2]     | 77.08      | 75.35    |
> |                 | **MLKD + TeKAP (Ours)**   | **77.36**  | **75.67**|
> |  **Multi-Teacher** | TAKD [3]      | 73.93      | 73.83    |
> |                 | **TAKD + TeKAP (Ours)**   | **74.81**  | **74.37**|
> |                 | CA-MKD [4] | 75.90      | 74.56    |
> |                 | **CA-MKD + TeKAP (Ours)**   | **76.34**  | **74.98**|
> |                 | DGKD [5]    | 75.31      | 74.23    |
> |                 | **DGKD + TeKAP (Ours)**   | **76.17**  | **75.14**|
>
>
> **Table-1:** The effects of TeKAP on the SOTA methods DKD [1], MLD [2], TAKD [3], CA-MKD [4], and DGKD [5].
>
> | #Original Teachers (T) | TeKAP (F+L) |
> |-------------------------|--------------|
> | 1 OriginT + 3 AugT             | 75.98        |
> | 2 OriginT + 3 AugT             | 76.12        |
> | 3 OriginT + 3 AugT             | 76.31        |
>
> Table 2: Effect of multiple original teachers.
>
>
> |Network |  Augmentation Techniques | TeKAP (F+L) |
> |-------------------------|--------------|--------------|
> |     **ResNet32x4-ResNet8x4**         | Gaussian             | 75.98        |
> |                                                             | Uniform             | 75.71        |
> |     **WRN_40_2-WRN_40_1**         | Gaussian             | 74.41        |
> |                                                             | Uniform             | 74.26        |
>
> Table 3: Effect of different noise techniques.
>
> ### Responses
> 1. **Theoretical Depth of Perturbation Methods:** We appreciate this insightful comment. We agree that the theoretical depth needs to improve. Actually, Gaussian noise was chosen for its simplicity and general applicability across various domains. However, we also show the effect of uniform distribution noise in Table 3. We have used zero mean and 1 std. to produce random noise on every epoch and perform a weighted combination with the original teacher logits (noise weights with 0.1 and teacher weights with 0.9). We also added detailed guidelines on how to set the noise parameters for different configurations in the revised manuscript.
>
>
> 2. **Comparison with SOTA Methods:** We agree that the evaluation could benefit from a broader scope. We have added comparisons with additional multi-teacher distillation methods in Table 1.
>
> 3. **Scalability and Computational Efficiency:** The training of a teacher ResNet32x4 in CIFAR100 using KD takes 16 seconds per epoch approximately. As we run for 240 epochs then the total time taken by is 240*16 = 64 minutes using 2, 3080 NVIDIA GeForce GPUs. For multi-teacher or ensemble learning we need to train multiple teachers, let's assume 2 teacher assistants of equal size which takes 64*2 = 128 minutes (approx) for DGKD. In our approach, TeKAP takes 18 seconds per epoch which is in total: 72 minutes only. We will add the complexity in the supplementary.
>
> 4. **Overclaimed Statements:**  Thank you very much for this insightful comment. We will improved the literature review in the final version.
>
> 5. **Gaussian Noise Parameters (Page 4):** We have used zero mean and 1 std. (standard Gaussian) to produce random noise on every epoch and perform a weighted combination with the original teacher logits (noise weights with 0.1 and teacher weights with 0.9). In future work, we will work on optimizing these hyperparameters.
>
>
> 6. **Incomplete Explanation of Terms (Page 5):** We have included the explanation in the revised manuscript.
>
> 7. **Overfitting Risk with Static Noise Parameters:** We agree with this comment. The static noise will create inductive bias shifts or over-fitting. This is why we generated random noise at every epoch which is considered as dynamic noise which creates diversity and balance. However, we are experimenting with static noise. The results will be reported here soon.
>
> 8. **Handling Class Imbalance:** We have included the results below **Update: Response to Reviewer pUYy (4)**.

---

> ### Author Response · Authors · 2024-11-24
> **Response to Reviewer pUYy (2/3)**
>
> 9. **Inter-Class Correlation Discussion (Page 8):** If two classes are strongly correlated in the teacher logits, random distortions will not eliminate this correlation but may perturb its exact magnitude or direction, leading to diverse interpretations of the relationship. Imagine teaching a concept by showing slightly varied examples, this helps learners generalize the concept rather than memorize specific instances. Similar to techniques like dropout (which can be considered implicitly network ensemble learning because every random dropping creates a different network structure), random feature distortion (considered as a diverse network as the outputs are slightly different so it is assumed they come from different networks) can force the model to adapt to a broader range of conditions. This diversity helps the student model avoid collapsing into a rigid interpretation of the teacher’s outputs.
>
>
> 10. **Number of Augmented Teachers (Figure 6):** Empirical results showed that three augmented teachers offer the optimal trade-off between diversity and stability. For future applications, we recommend using three augmented teachers as a default, balancing performance and computational cost. The observed performance with two teachers aligns with expectations due to reduced diversity compared to three.
>
>
> We are committed to incorporating these changes to strengthen the theoretical and experimental rigour of our work. The revised manuscript will provide a clearer understanding of TeKAP’s capabilities, limitations, and broader applicability.
> Thank you again for your valuable feedback.
>
> ### References
> 1. Zhao, Borui, et al. "Decoupled knowledge distillation." Proceedings of the IEEE/CVF Conference on computer vision and pattern recognition. 2022.
> 2. Jin, Ying, Jiaqi Wang, and Dahua Lin. "Multi-level logit distillation." Proceedings of the IEEE/CVF Conference on Computer Vision and Pattern Recognition. 2023.
> 3. Mirzadeh, Seyed Iman, et al. "Improved knowledge distillation via teacher assistant." Proceedings of the AAAI conference on artificial intelligence. Vol. 34. No. 04. 2020.
> 4. Zhang, Hailin, Defang Chen, and Can Wang. "Confidence-aware multi-teacher knowledge distillation." ICASSP 2022-2022 IEEE International Conference on Acoustics, Speech and Signal Processing (ICASSP). IEEE, 2022.
> 5. Son, Wonchul, et al. "Densely guided knowledge distillation using multiple teacher assistants." Proceedings of the IEEE/CVF International Conference on Computer Vision. 2021.

---

> ### Author Response · Authors · 2024-11-28
> **Response to Reviewer pUYy (3/3)**
>
> ### The improvements we have made:
> 1. **(Reviewers: NkEk, pUYy, zdP5, FGoU): Additional comparison with state-of-the-art:** Added to the revised manuscript (Table 2, page 7)
>
> 2. **(Reviewers: NkEk, pUYy, zdP5, FGoU) multi-teacher:** The results discussion for the recent SOTA multi-teacher approach is added to section 4.1, Table 2 (page 7) of the revised manuscript.
>
> 3. **(Reviewers: NkEk): explanation of usage scenarios between the feature level and logit level:** Added in section 3.1. Page 4 of the main manuscript. (Please find the changes marked highlights in the supplementary)
>
> 4. **(Reviewers: NkEk, pUYy) potential benefits of increasing the number of augmented teachers** Updated Figure 6 (Now Figure 5 of the main manuscript, Table 2 of this response). We have trained more teachers (till - 10) and provided the potential benefits of increasing the number of augmented teacher models in Table 1 of the supplementary.
>
> 5. **(Reviewers: NkEk) Evaluation of TeKAP on ensemble learning.** Added to the supplementary: Table 2, Section B. Table 3 of the last response.
>
> 6. **(Reviewer: pUYy): Theoretical Depth:** We have extended the theoretical analysis in the supplementary (Section K in details). more theoretical discussion in the supplementary (section D).
>
> 7. **(Reviewer: pUYy, FGoU, zdP5) effect for different Gaussian noise parameters:** We have used mean = 0 and variance = 1 as the default. Additionally, we added the effect for variance $\sigma$ = [0.5, 1, 1.5] in the supplementary (Table 5, section E).
>
> 8. **(Reviewer: pUYy) comparative computation complexity**: Added to section H of the supplementary.
>
> 9. **(Reviewer: pUYy, FGoU) Description and explanation of every mathematical term on page 5**: We have carefully gone through and added the description and explanation of every mathematical term used in the paper.
>
> 10. **(Reviewer: pUYy, FGoU) Experiments of the class imbalance data:** Added to the supplementary Table 4, section D.
>
> 11. **(Reviewer: pUYy, FGoU) fixed noise experiments**: Experiments are running and will be added to the final version and we will also report here with the deadline.
>
> 12. **(Reviewer: pUYy. zdP5) how inter-class diversity works**: Discussion added in the supplementary section I.
>
> 13. **(Reviewer: zdP5) effect for different values of $\lambda$**: Added in the supplementary Table 3, Section C.
>
> 14. **(Reviewer: zdP5) Meaning of $L_{cel}:** We have added the meaning of $L_{cel}$ in line 209, page 5 of the main manuscript.
>
> 15. **(Reviewer: zdP5) More experiments on TAKD with WRN-22-2 or WRN-16-2?**: Experiments are running. We will be added in the final version and report here soon.
>
> 16. **(Reviewer: FGoU) clarification of random distortion and inter-class relationships**: Added in the supplementary in section I.
>
>
> Thank you for taking the time to provide detailed and thoughtful comments. Your feedback has been instrumental in improving the manuscript. We are deeply grateful for your insights and are ready to respond to any further questions or concerns.

---

> ### Author Response · Authors · 2024-11-29
> **Update: Response to Reviewer pUYy (4)**
>
> We have added new results on class imbalance dataset, effects of different values for variance $\sigma$ for Gaussian, and various valus of $\gamma$.
>
> ***Results-A: Class Imbalance Dataset:**
>
> | Methods | ResNet32x4-ResNet8x4 | WRN_40_2-WRN_16_2 | VGG13-VGG8 |
> |-------------------------|--------------|---------------|------------------|
> | Baseline (KD) | 41.71 | 52.08 | 47.52 |
> | + TeKAP (Ours) | 46.42 | 52.72 | 51.25 |
>
> Table 5: Significance of TeKAP on class imbalance dataset.  We have used the class distribution of the CIFAR100 dataset that is described in Table 6 (of the supplementary)
>
>
> Section D (of the supplementary)
> The results presented in Table 5 (4 of the supplementary) highlight the effectiveness of TeKAP in addressing class imbalance in knowledge distillation tasks. TeKAP consistently improves the performance of all three teacher-student model pairs (ResNet32x4-ResNet8x4, WRN\_40\_2-WRN\_16\_2, and VGG13-VGG8) compared to the baseline Knowledge Distillation (KD) approach. Specifically, TeKAP boosts accuracy by 4.71\% for ResNet32x4-ResNet8x4, 0.64\% for WRN\_40\_2-WRN\_16\_2, and 3.73\% for VGG13-VGG8. These results indicate that TeKAP is particularly effective in enhancing performance for models with lower baseline accuracy, though it also provides improvements for models with higher baseline accuracy. This suggests that TeKAP can effectively mitigate the effects of class imbalance, leading to improved generalization in knowledge distillation tasks.
>
> **Results-B: Effect of TeKAP for different variance $\sigma$ on the performance:**
>
> **Table 6**: Effect of TeKAP with different variance $\sigma$. KD is used as the baseline distillation approach. We have used mean zero in all the cases.
>
> | Variance      | $\sigma = 0.5$ | $\sigma = 1$ | $\sigma = 1.5$ |
> |---------------|----------------|--------------|----------------|
> | Accuracy      | 74.89          | 74.79        | 74.35          |
>
> Section E (of the supplementary)
>
> Table 6 (table 5 of the supplementary and section E) summarizes the impact of different variances ($\sigma$) on the performance of TeKAP, using the CIFAR-100 dataset. The baseline distillation approach, Knowledge Distillation (KD), is used for comparison. As shown in the results, the accuracy of the model remains relatively stable across varying values of $\sigma$. Specifically, when $\sigma = 0.5$, the model achieves an accuracy of 74.89\%, slightly higher than the accuracy at $\sigma = 1$ (74.79\%) and $\sigma = 1.5$ (74.35\%). These results suggest that, within the range of variances tested, increasing the noise variance does not significantly degrade performance. In fact, the accuracy only decreases marginally as the variance increases from 0.5 to 1.5, which indicates the robustness of TeKAP with respect to noise. This behavior suggests that TeKAP can maintain competitive performance even with varying levels of noise in the teacher models, highlighting its resilience to noise during distillation. The consistent results across different variances also support the idea that TeKAP is stable and less sensitive to slight perturbations in the teacher’s logits. This stability is critical for practical applications where noise may be present in the data or models.
>
>
> ***Results-C: Effect of various $\lambda$:**
>
> Table 3 (of the supplementary): Effect of the different values of λ (the weights of the noise terms). AugT denotes augmented teachers.
>
> | Number of AugT | λ = 0.2 | λ = 0.4 | λ = 0.6 | λ = 0.8 |
> |----------------|---------|---------|---------|---------|
> | AugT = 5       | 74.26   | 74.46   | 74.63   | 75.12   |
> | AugT = 10      | 74.29   | 74.73   | 74.85   | 74.98   |
>
> ### Section C of the supplementary
>
> The results in Table  3 (of the supplementary) demonstrate the effect of varying the noise weight ($\lambda$) and the number of augmented teachers (AugT) on the performance of the student model. For AugT=5, the accuracy consistently improves as $\lambda$ increases, starting from $74.26\%$ at $\lambda=0.2$ and reaching $75.12\%$ at $\lambda=0.8$. This trend indicates that higher noise weights contribute positively to the student’s generalization by introducing greater diversity. Similarly, for AugT=10, the performance improves from $74.29\%$ at $\lambda=0.2$ to $74.98\%$ at $\lambda=0.8$, but the gains are less pronounced compared to AugT=5, suggesting a saturation effect with a larger number of augmented teachers.

---

> ### Author Response · Authors · 2024-11-29
> **Update: Response to Reviewer pUYy (5):**
>
> ### Result D Static (Fixed) vs Dynamic Noise (We will add this response to the supplementary of the final version)
>
> **Results D: Additional Experiments**
>
> Table D:  Evaluation of the comparative effects between static and dynamic noise. KD has been used as the baseline distillation approach. The experiment is conducted with three augmented teachers. We use $\sigma = 1$, $\lambda = 0.8$, and three augmented teachers. Gaussian noise is used to generate the noise.
>
> | **Methods**               | ResNet32x4-ResNet8x4 | WRN\_40\_2-WRN\_16\_2 | VGG13-VGG8 |
> |---------------------------|----------------------|-----------------------|------------|
> | **Baseline (KD)**         | 73.33               | 74.92                | 72.98      |
> | **+ TeKAP (Static-L)**    | 73.74               | 74.66                | 73.29      |
> | **+ TeKAP (Ours: Dynamic-L)** | 74.79           | 75.21                | 74.00      |
>
> The results in Table C demonstrate the effectiveness of dynamic noise over static noise and the baseline Knowledge Distillation (KD) approach across three teacher-student pairs: ResNet32x4-ResNet8x4, WRN\_40\_2-WRN\_16\_2, and VGG13-VGG8 on CIFAR100 dataset. Baseline KD provides solid performance, achieving 73.33\%, 74.92\%, and 72.98\% accuracy, respectively. Incorporating static noise (TeKAP Static-L) shows minor improvements for ResNet32x4-ResNet8x4 and VGG13-VGG8, achieving 73.74\% and 73.29\%, but performs slightly worse (74.66\%) for WRN\_40\_2-WRN\_16\_2, indicating its limited adaptability. Conversely, our proposed dynamic noise strategy (TeKAP Dynamic-L) consistently outperforms both static noise and baseline KD, achieving significant gains with accuracies of 74.79\%, 75.21\%, and 74.00\%, respectively. This superiority stems from dynamic noise's adaptability enabling robust generalization. These findings underscore the robustness and efficacy of dynamic noise in enhancing knowledge transfer during distillation, providing a compelling case for its application in improving student network performance across diverse architectures.
>
>
> **We will add this response (ablation study: the comparative effect between static vs dynamic noise on TeKAP in the supplementary).**
>
> Thank you for your detailed and insightful comments. Your feedback has significantly contributed to improving the manuscript. We are happy to address any additional questions or concerns you may have.

---

> > ### Author Response · Authors · 2024-12-02
> > **Gentle Reminder with Gratitude**
> >
> > Dear Reviewer,
> >
> > We wish to convey our heartfelt gratitude and appreciation for your insightful and constructive feedback. As the discussion period is anticipated to conclude very soon, we kindly request you to share any additional questions or concerns you may have.
> > We remain readily available and would be pleased to continue the dialogue to ensure that all matters are comprehensively addressed.

---

> ### Author Response · Authors · 2024-12-03
>
> Dear Reviewer pUYy,
>
> Thank you for your thoughtful comments and suggestions. We have added further results with detailed, point-by-point explanations for your review. We hope these enhancements meet your expectations, and if you feel they merit reconsidering your rating, we would be truly grateful.
>
> Best regards,
> The Authors

---

### Official Review · Reviewer_NkEk · 2024-11-08

**Soundness:** 2
**Presentation:** 4
**Contribution:** 3
**Rating:** 6
**Confidence:** 3

**Summary:**

The paper proposes a novel knowledge distillation method called TeKAP (Teacher Knowledge Augmentation via Perturbation), which generates diverse perspectives from a single teacher model. Instead of relying on multiple teacher models for supervision, TeKAP introduces diversity by perturbing both feature maps and output logits of a pretrained teacher network. This approach aims to simulate the benefits of multi-teacher distillation without the associated computational cost.

**Strengths:**

- The paper provides thorough theoretical proof and experimental validation.
- The paper is well-structured and clear in its approach, with intriguing perspectives.
- The method proposed in the paper has a wide range of application scenarios.

**Weaknesses:**

- There is a lack of comparison with recent multi-teacher distillation work.
- The explanation of the difference in usage scenarios between feature-level and logit-level may be insufficient..

**Questions:**

- If more distillation methods could be included, it would be more convincing.
- I think the idea that different teacher models provide different perspectives is interesting. Would increasing the number of teacher models further improve performance?

---

> ### Author Response · Authors · 2024-11-24
> **Response to Reviewer NkEk**
>
> We thank you for the valuable feedback and suggestions on our submission. We have addressed the comments and questions step-by-step below:
>
> ### **Additional Experiments:**
>
> |        | Model         | ResNet32x4-ResNet8x4 | WRN_40_2-WRN_40_1 |
> |-----------------|---------------|------------|----------|
> | **Teacher**     | Accuracy      | 79.42      | 75.61    |
> | **Student**     | Accuracy      | 72.50      | 71.98    |
> | **Single Teacher** | DKD [1]   | 76.32      | 74.81    |
> |                 | **DKD + TeKAP (Ours)**   | **76.59**  | **75.33**|
> |                 | MLKD [2]     | 77.08      | 75.35    |
> |                 | **MLKD + TeKAP (Ours)**   | **77.36**  | **75.67**|
> |  **Multi-Teacher** | TAKD [3]      | 73.93      | 73.83    |
> |                 | **TAKD + TeKAP (Ours)**   | **74.81**  | **74.37**|
> |                 | CA-MKD [4] | 75.90      | 74.56    |
> |                 | **CA-MKD + TeKAP (Ours)**   | **76.34**  | **74.98**|
> |                 | DGKD [5]    | 75.31      | 74.23    |
> |                 | **DGKD + TeKAP (Ours)**   | **76.17**  | **75.14**|
>
>
> **Table-1:** The effects of TeKAP on the SOTA methods DKD [1], MLKD [2], TAKD [3], CA-MKD [4], and DGKD [5].
>
>
> ###  **Responses**:
>
> 1. **Weakness-1: More comparison with recent multi-teacher work:** We have included more comparisons with DKD, MLKD, TAKD, CA-MKD, and DGKD, evaluated on the CIFAR-100 dataset. TeKAP outperformed all approaches under every scenario.
>
> 2. **Weakness-2: Insufficient explanation of the difference in usage scenarios between feature-level and logit-level:** Thanks for this insightful suggestion. Logit-level augmentation primarily diversifies the inter-class relationships, providing alternative supervisory signals that regularize the student network. Feature-level augmentation, on the other hand, introduces diversity in intermediate feature representations, exposing the student to a broader spectrum of variations (like dropout or data augmentation). Both augmentations target distinct aspects of teacher knowledge: logits focus on prediction diversity, while features address internal representation diversity.
>
> 3. **Question-1: Inclusion of more distillation methods for a more convincing study:** We appreciate the suggestion. In our revised manuscript, We will add more comparisons with SOTA techniques as shown in Table 1 (here).
>
> | #Teachers | Ours | Baseline (KD) | Baseline (Rerun) |
> |-------------------------|--------------|---------------|------------------|
> | T + 1 AugT             | 73.9         | 72.98         | 73.3            |
> | T + 2 AugT             | 73.43        | 72.98         | 73.3            |
> | T + 3 AugT             | 74.04        | 72.98         | 73.3            |
> | T + 4 AugT             | 73.98        | 72.98         | 73.3            |
> | T + 5 AugT             | 74.00        | 72.98         | 73.3            |
> | T + 6 AugT             | 73.53        | 72.98         | 73.3            |
> | T + 7 AugT             | 74.16        | 72.98         | 73.3            |
> | T + 8 AugT             | 74.33        | 72.98         | 73.3            |
> | T + 9 AugT             | 74.63        | 72.98         | 73.3            |
> | T + 10 AugT            | 75.11         | 72.98         | 73.3            |
>
> Table 2: Effect of number of the augmented teachers.
>
> 4. **Question-2: Potential benefits of increasing the number of teacher models:**  In Figure 6, we show that TeKAP consistently benefits from additional synthetic teachers up to a certain threshold. Increasing the number of augmented teacher models will not further improve the performance because too much noise will destroy the teacher knowledge pattern instead of regularizing (similar to dropout or data augmentation). The same thing happens in ensemble learning where excessive models may introduce noise. We have run additional experiments till 10 augmented teachers.
>
> | #Original Teachers (T) | TeKAP (Ours) |
> |-------------------------|--------------|
> | 1 OriginT + 3 AugT             | 75.98        |
> | 2 OriginT + 3 AugT             | 76.12        |
> | 3 OriginT + 3 AugT             | 76.31        |
>
> Table 3: Effect of multiple original teachers.
>
> 5. **Inspired By: More teachers with augmentation of every teacher:** We have run additional experiments where we use multiple teachers (2 and 3) of ResNet32x4 (training using different seeds and lr). We have augmented each teacher with three noise sets. We experience performance improvements while increasing the number of teachers as shown in Table 3.
>
> Again, thank you very much for these insightful suggestions. We experience that these suggestions help improve the script. We will add these responses to our revised manuscript accordingly.

---

> ### Author Response · Authors · 2024-11-25
> **References**
>
> ### References
> 1. Zhao, Borui, et al. "Decoupled knowledge distillation." Proceedings of the IEEE/CVF Conference on computer vision and pattern recognition. 2022.
> 2. Jin, Ying, Jiaqi Wang, and Dahua Lin. "Multi-level logit distillation." Proceedings of the IEEE/CVF Conference on Computer Vision and Pattern Recognition. 2023.
> 3. Mirzadeh, Seyed Iman, et al. "Improved knowledge distillation via teacher assistant." Proceedings of the AAAI conference on artificial intelligence. Vol. 34. No. 04. 2020.
> 4. Zhang, Hailin, Defang Chen, and Can Wang. "Confidence-aware multi-teacher knowledge distillation." ICASSP 2022-2022 IEEE International Conference on Acoustics, Speech and Signal Processing (ICASSP). IEEE, 2022.
> 5. Son, Wonchul, et al. "Densely guided knowledge distillation using multiple teacher assistants." Proceedings of the IEEE/CVF International Conference on Computer Vision. 2021.

---

> > ### Comment · Reviewer_NkEk · 2024-11-26
> >
> > Thanks to the author's reply. I would like to keep my score.

---

> ### Author Response · Authors · 2024-11-28
> **Further Additional Responses to Reviewer NkEk (1/2)**
>
> We appreciate the valuable time and efforts offered by the Reviewer NkEk. Please note that we have updated the results in Table 2 of the previous response. We have added more results and a details analysis of our paper based on concerns raised by all the reviewers.
>
> We would like to request to have a look again at our revised manuscript and overall responses. We would be happy if the reviewer again go through the responses, revised manuscript, supplementary, and reassess our updated manuscript.
>
> The updated Table 2 can be found as:
> **Correction: Table 2**
>
>  The reported result was confused with the class-imbalanced experiments. However, inspired by these comments we carefully went through again the experiments and evaluations. **We have updated Table 2 of the previous responses**. The updated results are reported in Figure 5 and Section 4.9 of the revised manuscript.
>
>
> | #Teachers | Ours | Baseline (KD) | Baseline (Rerun) |
> |-------------------------|--------------|---------------|------------------|
> | T + 1 AugT             | 73.9         | 72.98         | 73.3            |
> | T + 2 AugT             | 73.43        | 72.98         | 73.3            |
> | T + 3 AugT             | 74.04        | 72.98         | 73.3            |
> | T + 4 AugT             | 73.98        | 72.98         | 73.3            |
> | T + 5 AugT             | 74.00        | 72.98         | 73.3            |
> | T + 6 AugT             | 73.53        | 72.98         | 73.3            |
> | T + 7 AugT             | 74.16        | 72.98         | 73.3            |
> | T + 8 AugT             | 74.33        | 72.98         | 73.3            |
> | T + 9 AugT             | 74.63        | 72.98         | 73.3            |
> | T + 10 AugT            | 75.11         | 72.98         | 73.3            |
>
> Table 4: Effect of number of the augmented teachers.
>
> Table. 4 (Fig. 5 of the main manuscript) shows the effect of the number of augmented teachers. We use ResNet32x4-ResNet8x4 as the teacher-student setups on the CIFAR100 dataset to examine the effect of the hyper-parameters. From Table.4 (Fig. 5 of the main manuscript) we see that TeKAP is robust to the number of augmented teachers. For every number of augmented teachers, TeKAP achieves better accuracy than baseline and DKD students. The best performance is achieved when the number of the augmented teacher is $3$. We have used three ($3$), and one $(1)$ augmented teacher along with the original teacher, respectively. During feature and logit distortion, the weights for noise and teacher output are $0.1$, and $0.9$, respectively.
>
> ### Concerns
> We have updated our manuscript based on the reviews added supplementary documents, and revised the manuscript with changes highlights. The updated manuscript's clean version, changes marked with yellow, and supplementary are available now.
>
> ### The improvements we have made:
> 1. **(Reviewers: NkEk, pUYy, zdP5, FGoU): Additional comparison with state-of-the-art:** Added to the revised manuscript (Table 2, page 7)
>
> 2. **(Reviewers: NkEk, pUYy, zdP5, FGoU) multi-teacher:** The results discussion for the recent SOTA multi-teacher approach is added to section 4.1, Table 2 (page 7) of the revised manuscript.
>
> 3. **(Reviewers: NkEk): explanation of usage scenarios between the feature level and logit level:** Added in section 3.1. Page 4 of the main manuscript. (Please find the changes marked highlights in the supplementary)
>
> 4. **(Reviewers: NkEk, pUYy) potential benefits of increasing the number of augmented teachers** Updated Figure 6 (Now Figure 5 of the main manuscript, Table 2 of this response). We have trained more teachers (till - 10) and provided the potential benefits of increasing the number of augmented teacher models in Table 1 of the supplementary.
>
> 5. **(Reviewers: NkEk) Evaluation of TeKAP on ensemble learning.** Added to the supplementary: Table 2, Section B. Table 3 of the last response.
>
> 6. **(Reviewer: pUYy): Theoretical Depth:** We have extended the theoretical analysis in the supplementary (Section K in details). more theoretical discussion in the supplementary (section D).
>
> 7. **(Reviewer: pUYy, FGoU, zdP5) effect for different Gaussian noise parameters:** We have used mean = 0 and variance = 1 as the default. Additionally, we added the effect for variance $\sigma$ = [0.5, 1, 1.5] in the supplementary (Table 5, section E).
>
> 8. **(Reviewer: pUYy) comparative computation complexity**: Added to section H of the supplementary.
>
> 9. **(Reviewer: pUYy, FGoU) Description and explanation of every mathematical term on page 5**: We have carefully gone through and added the description and explanation of every mathematical term used in the paper.
>
> 10. **(Reviewer: pUYy, FGoU) Experiments of the class imbalance data:** Added to the supplementary Table 4, section D.

---

> ### Author Response · Authors · 2024-11-28
> **Further Additional Responses to Reviewer NkEk (2/2)**
>
> 11. **(Reviewer: pUYy, FGoU) fixed noise experiments**: Experiments are running and will be added to the final version and we will also report here with the deadline.
>
> 12. **(Reviewer: pUYy. zdP5) how inter-class diversity works**: Discussion added in the supplementary section I.
>
> 13. **(Reviewer: zdP5) effect for different values of $\lambda$**: Added in the supplementary Table 3, Section C.
>
> 14. **(Reviewer: zdP5) Meaning of $L_{cel}:** We have added the meaning of $L_{cel}$ in line 209, page 5 of the main manuscript.
>
> 15. **(Reviewer: zdP5) More experiments on TAKD with WRN-22-2 or WRN-16-2?**: Experiments are running. We will be added in the final version and report here soon.
>
> 16. **(Reviewer: FGoU) clarification of random distortion and inter-class relationships**: Added in the supplementary in section I.
>
> We appreciate the effort you put into reviewing our manuscript. Your suggestions have been invaluable in refining and improving its quality. Thank you for your thoughtful comments, and we would be glad to address any further queries.

---

> > ### Author Response · Authors · 2024-12-03
> >
> > Dear Reviewer NkEk,
> >
> > We greatly value your encouraging feedback and recognition of our work and contributions.
> >
> > Additionally, we have included further results and kindly request you to review them at your convenience. We appreciate your time and thoughtful consideration.
> >
> > Best regards,
> > The Authors

---

### Meta-Review · Area_Chair_aKQy · 2024-12-20

**Metareview:**

This paper introduces TeKAP, a novel knowledge distillation method that generates diverse teacher perspectives by perturbing the feature maps and logits of a single pretrained teacher model. It simulates the benefits of multi-teacher distillation while reducing computational cost, improving student model generalization, and demonstrating effectiveness on standard benchmarks.

The paper has received mixed scores: three weak positives (6, 6, 6) and one negative (5). The reviewers highlight some its strengths:

(1). Efficiency and Simplicity: The method uses a single pretrained teacher to simulate multiple teacher perspectives through perturbation, effectively circumventing the high computational costs of traditional multi-teacher setups.

(2). Seamless Integration with Existing KD Methods: The plug-and-play module integrates well with existing knowledge distillation methods, adding minimal computational overhead.

(3). Wide Range of Applications: The proposed method demonstrates promising results in various aspects, such as model compression, adversarial robustness, transferability, and few-shot learning, indicating its broad applicability.

(4). Clear Structure and Expression: The paper is well-structured, clear, and easy to follow, presenting intriguing perspectives.

Meanwhile, the reviewers also pointed out some key weaknesses of the paper, such as the lack of comparisons with recent multi-teacher distillation approaches and other state-of-the-art single-teacher methods, which makes it difficult to highlight the relative strengths of the proposed method. The paper also has insufficient theoretical analysis of perturbation methods and lacks sufficient details on implementation. After the rebuttal phase, most of the weaknesses have been addressed.

The final decision is acceptance based on the following primary reasons: method's efficiency and simplicity, and seamless integration with existing KD methods to improve performance. Besides, most weaknesses have been addressed after the rebuttal phase. The authors are required to include some of the improvements mentioned in the rebuttal, such as necessary experimental results and image enhancements, in the final version. Meanwhile, the authors should consider the reviewers' suggestions to further improve the quality of the final version of the paper.

**Additional Comments On Reviewer Discussion:**

The paper has received mixed scores: three weak positives (6, 6, 6) and one negative (5).

Reviewer NkEk praised the theoretical proof, experimental validation, and broad applicability of the method but suggested clarifying feature-level vs logit-level scenarios.

Reviewer pUYy highlighted the novelty of using a single pretrained teacher and the promising results but raised concerns about the lack of theoretical depth, missing comparisons with other methods, and no discussion on computational efficiency.

Reviewer zdP5 appreciated the efficiency of the method.

Reviewer FGoU valued the simplicity of the plug-and-play module but requested more details on the noise perturbation implementation.

Most reviewers expressed concerns about the lack of comparisons with state-of-the-art methods and the limited validation of the proposed approach. Additionally, several reviewers highlighted the need for more detailed discussions on the theoretical aspects and method's implementation details, such as perturbation parameters and computational efficiency. After the rebuttal phase, most of the concerns have been addressed.

AC believes that this paper indeed proposes a novel knowledge distillation method and provides a substantial amount of experimental data to demonstrate the effectiveness of their approach. Additionally, the authors have provided reproducible open-source code. Therefore, AC is inclined to accept the paper. Based on these considerations, the final decision is accept.

---

### Decision · Program_Chairs · 2025-01-22

Accept (Poster)